# Optimal Treatment Allocation for Efficient Policy Evaluation in Sequential Decision Making

**Ting Li**[1]    **Chengchun Shi**[2]    **Jianing Wang**[1]    **Fan Zhou**[1]    **Hongtu Zhu**[3] [*]

[1]School of Statistics and Management, Shanghai University of Finance and Economics
[2]Department of Statistics, London School of Economics and Political Science
[3]Department of Biostatistics, University of North Carolina at Chapel Hill
`tingli@mail.shufe.edu.cn, c.shi7@lse.ac.uk, jianing.wang@163.sufe.edu.cn`
`zhoufan@mail.shufe.edu.cn,htzhu@email.unc.edu`

## Abstract

A/B testing is critical for modern technological companies to evaluate the effectiveness of newly developed products against standard baselines. This paper studies optimal designs that aim to maximize the amount of information obtained from online experiments to estimate treatment effects accurately. We propose three optimal allocation strategies in a dynamic setting where treatments are sequentially assigned over time. These strategies are designed to minimize the variance of the treatment effect estimator when data follow a non-Markov decision process or a (time-varying) Markov decision process. We further develop estimation procedures based on existing off-policy evaluation (OPE) methods and conduct extensive experiments in various environments to demonstrate the effectiveness of the proposed methodologies. In theory, we prove the optimality of the proposed treatment allocation design and establish upper bounds for the mean squared errors of the resulting treatment effect estimators.

## 1  Introduction

**Motivation**. Prior to the full-scale deployment of any product in practical applications, an accurate evaluation of its potential impact is crucial. Modern technology companies, including Amazon, Google, Netflix, Uber, and Didi, commonly employ online experimentation or A/B testing as a means of evaluating the effectiveness of new products or policies (treatment) in comparison to their existing counterparts (control). Often, in these experiments, policies are assigned sequentially over time, impacting both current and future responses. Such a dynamic can invalidate the Stable Unit Treatment Value Assumption (SUTVA), as outlined by (Imbens and Rubin, 2015). Failure to consider these temporal carryover effects can lead to a biased estimation of the treatment effect. Additionally, the limited duration of the experiment and the typically small size of the treatment effect pose significant challenges to consistently detecting the treatment effect.

A substantial body of literature has been dedicated to developing policy evaluation or A/B testing algorithms using data from online experiments. However, there has been limited focus on the generation of the experimental dataset itself. This factor is critical as it can substantially influence the precision of the subsequent treatment effect estimator. This paper's primary focus is to study optimal experimental designs in the context of sequential decision making. In clinical trials, a carefully designed experiment can significantly improve the accuracy of the treatment effect estimator and the statistical power of the associated test, as noted by (Sverdlov et al., 2020). However, most existing

---

[*]The first two authors contribute equally to this paper. Address for correspondence: Hongtu Zhu, Ph.D., E-mail: htzhu@email.unc.edu.

37th Conference on Neural Information Processing Systems (NeurIPS 2023).

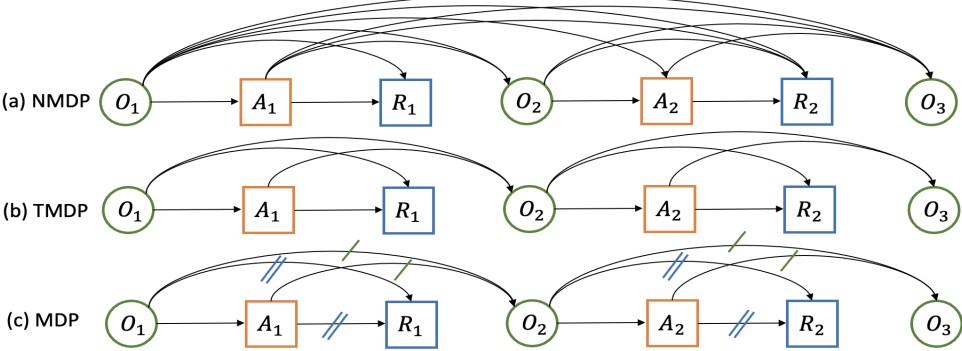

Figure 1: Data structure of NMDP, TMDP, and MDP. (a) In the NMDP, the reward and future observations are determined by all past observation-action pairs. (b) In the TMDP, the reward and future observations depend solely on the current observation-action pairs. (c) In the MDP, the reward and future observations also rely on the current observation-action pairs, with the same-colored slashes indicating identical conditional distributions.

designs consider the classic identically and independently distributed (i.i.d.) setting, failing to account for temporal carryover effects.

**Contributions**. We make three important contributions below. Firstly, we propose three optimal dynamic treatment allocation strategies in sequential decision making, while accounting for carryover effects over time. The proposed designs are built upon the $A$-optimality criterion developed in non-dynamic settings (Atkinson et al., 2007), and effectively minimize the variance of the average treatment effect estimator when data follow a (time-invariant) Markov decision process (MDP), a time-varying MDP (TMDP) or a non-Markov decision process (NMDP). Figure 1 offers a visual illustration of these data generating processes. To our knowledge, this represents the first piece of work to provide an analytical form of optimal allocation strategies within these dynamic contexts.

Secondly, by leveraging insights from the established field of off-policy evaluation (OPE, see, for instance, Dudík et al., 2014; Uehara et al., 2022, for reviews), we devise subsequent estimation methodologies that learn the treatment effect estimator using log data produced in accordance with our proposed design. We employ real datasets gathered from a globally recognized ride-sharing company to create a city-scale synthetic environment that mimics the spatio-temporal dynamics of the order dispatching problem. Our findings indicate that the proposed estimators deliver significantly superior accuracy compared to conventional baseline estimators.

Thirdly, we establish the theoretical properties of the proposed designs and estimators. Specifically, we establish that in NMDPs, the optimal design distributes the initial policy randomly, with probabilities corresponding to the standard deviations of the cumulative rewards, and subsequently implements the same policy. However, such a design may not necessarily be optimal in (T)MDPs. To address this, we focus on a category of restricted designs and identify sufficient conditions that ascertain their asymptotic optimalities in (T)MDPs. Moreover, we derive an upper limit for the mean squared error (MSE) of the resulting treatment effect estimator. Extensive simulation studies have been conducted to assess the finite sample performance of the proposed algorithms. Python code implementing the proposed algorithms is available at `https://github.com/tingstat/MDP_design`.

## 1.1 Related work

**Experimental designs**. There is an extensive body of literature on experimental design for clinical trials, with a multitude of optimal designs proposed. These include the global treatment balance design (Efron, 1971; Wei, 1977), designs that regulate the marginal covariate distribution across various treatment groups (Pocock and Simon, 1975; Heritier et al., 2005; Liu and Hu, 2022), and criteria-based design anchored on variance minimization (Begg and Iglewicz, 1980; Wong and Zhu, 2008; Chandereng et al., 2020). Other noteworthy designs are the $D$-optimality and its generalization $D_A$ optimality (Jones and Goos, 2009; Atkinson and Pedrosa, 2017; Loux, 2013; Liu et al., 2022), alongside the $A$-optimality and its extension $A_A$ optimality (Atkinson et al., 2007; Sverdlov and Rosenberger, 2013; Yin and Zhou, 2017). Moreover, numerous sequential adaptive designs for clinical trials have been developed, including covariate-adaptive designs (Zhu and Hu, 2019; Wang

and Ma, 2021), response-adaptive designs (Zhu et al., 2020; Yu et al., 2022), and covariate-adjusted response-adaptive designs (Zhang et al., 2007; Villar and Rosenberger, 2018). However, these methods were designed for i.i.d. data and thus are not directly applicable to our settings.

Recently, several papers have studied experimental designs with spatial/network spillover effects (Ugander et al., 2013; Baird et al., 2018; Li et al., 2019; Jagadeesan et al., 2020; Nandy et al., 2020; Viviano, 2020; Zhou et al., 2020; Kong et al., 2021; Leung, 2022). Additionally, a few designs have been developed for modern technological companies, such as e-commerce (Bajari et al., 2021; Nandy et al., 2021) and ride-sharing (Johari et al., 2022). However, these studies differ from ours in that they did not utilize NMDP or TMDP models for experimental designs.

Bojinov et al. (2023) explored optimal regular switchback design in the presence of a single experimental unit. Their approach relies on the assumption that the carryover effects last for a fixed duration. However, this assumption is not consistent with our NMDP and (T)MDP models, where the carryover effects, due to state transitions, can potentially last indefinitely.

Finally, Bayesian Optimal Experimental Design (BOED) presents a powerful tool for numerically determining the optimal design (see e.g., Ryan et al., 2016; Rainforth et al., 2023). Recently, works such as Foster et al. (2021), Lim et al. (2022), and Blau et al. (2022) have proposed leveraging deep reinforcement learning to compute the BOED. In contrast, our paper employs a frequentist approach, analytically deriving optimal designs for sequential decision making.

**Off-policy evaluation**. Our work is closely related to OPE, which seeks to estimate the mean outcome of a new target policy using logged data collected by a different policy. It has been applied in a range of domains including healthcare (Murphy, 2003), education (Mandel et al., 2014), ride-sharing (Shi et al., 2022) and video-sharing (Xu et al., 2023). Recently, it has been employed to conduct A/B testing and mediation analysis in the presence of temporal carryover effects (Hu and Wager, 2022; Farias et al., 2022; Shi et al., 2022; Tang et al., 2022; Ge et al., 2023; Shi et al., 2023). Existing algorithms include value-based methods (Mannor et al., 2007; Antos et al., 2008; Le et al., 2019; Feng et al., 2020; Luckett et al., 2020; Hao et al., 2021; Liao et al., 2021; Chen and Qi, 2022; Shi et al., 2022; Bian et al., 2023), importance (re)sampling-based methods (Swaminathan and Joachims, 2015; Liu et al., 2018; Schlegel et al., 2019; Yin and Wang, 2020; Thams et al., 2021; Wang et al., 2021; Hu and Wager, 2023), and doubly robust or more robust methods (Zhang et al., 2013; Jiang and Li, 2016; Thomas and Brunskill, 2016; Farajtabar et al., 2018; Kallus and Uehara, 2020; Uehara et al., 2020; Shi et al., 2021; Kallus and Uehara, 2022; Liao et al., 2022; Xie et al., 2023). In particular, Kallus and Uehara (2022) and Liao et al. (2022) established the efficiency bounds for OPE, which we leverage to construct optimal designs. However, none of the previous work considered how to design the behavior policy that generates high-quality data to improve the efficiency of the policy value estimator.

Recently, Wan et al. (2022) studied safe experimental designs for efficient policy evaluation under certain safety constraints in a non-dynamic setting without temporal carryover effects. However, it remains unclear how to generalize their methodologies to sequential decision making.

**A/B testing**. Lastly, our proposal aligns with a body of literature that develops A/B testing algorithms for randomized online experiments (see e.g., Kharitonov et al., 2015; Johari et al., 2017; Yang et al., 2017; Bojinov and Shephard, 2019; Ju et al., 2019; Zhou et al., 2020; Shi et al., 2021; Maharaj et al., 2023; Wan et al., 2023). However, most works focus on the sequential monitoring problem, which involves partitioning the significance level at each interim stage to control the overall type-I error. This is distinct from our goal, which is to generate high-quality experimental data to enhance the efficiency of estimation and inference.

## 2  Models and Problem Formulation

**Data**. We consider an experiment spanning $n$ days and each day is divided into $T$ time intervals. On the $i$-th day ($i = 1, \ldots, n$) and time interval $t$ ($t = 1, \ldots, T$), the decision maker observes certain time-varying market features, denoted as $O_t^{(i)}$ (e.g., the numbers of incoming orders and available drivers in a ride-sharing platform) and chooses to implement one of the two policies. This action is denoted as $A_t^{(i)} \in \{0, 1\}$. By convention, $A_t^{(i)} = 1$ indicates the implementation of a new policy, while $A_t^{(i)} = 0$ signifies the use of the control policy. Afterwards, the decision maker receives an immediate reward $R_t^{(i)}$ and subsequently observes the next observation $O_{t+1}^{(i)}$. This process is

repeated until we reach the termination time $T$. The observed data can thus be summarized as $\{O_t^{(i)}, A_t^{(i)}, R_t^{(i)}, t = 1, \ldots, T\}_{i=1}^n$.

**Model**. We consider three models for the aforementioned data generating process: an NMDP, a TMDP, and a (time-invariant) MDP. As depicted in Figure 1, an NMDP is a versatile model that doesn't impose Markovian or stationarity assumptions. The immediate reward ($R_t$) and future observation ($O_{t+1}$) depend on the entire history of observations and actions, not merely the current observation-action pair ($O_t$ and $A_t$). This kind of model has been extensively used in research on learning optimal dynamic treatment regimes (Kosorok and Laber, 2019; Tsiatis et al., 2019). Conversely, a TMDP is a special case of NMDP, with a stronger structural assumption that satisfies Markovanity. It assumes that the current observation-action pair is sufficient to determine the immediate reward and future observation, while allowing this conditional distribution function to vary with time. This type of data generating process is also known as an episodic MDP and has been widely studied in the reinforcement learning literature (see e.g., Jin et al., 2018, 2021; Li et al., 2023). Finally, an MDP is a special TMDP with an additional stationarity assumption.

**Objective**. A history-dependent policy, denoted as $\pi$, is a sequence of decision rules $\{\pi\}_{t \geq 1}$. Each rule, $\pi_t(\bullet|H_t)$, uses the observed data history $H_t$ up to time $t$ as input and provides a probability distribution as output. This output determines the likelihood of choosing each action $A_t$ at time $t$. The primary objective here is to estimate the Average Treatment Effect (ATE), which is defined as:

$$\text{ATE} = T^{-1} \sum_{t=1}^T [\mathbb{E}^1(R_t) - \mathbb{E}^0(R_t)],$$

where $\mathbb{E}^1$ (or $\mathbb{E}^0$) signifies the expected value when the system follows the new (or control) policy. Moreover, for any $a$ in the set $\{0, 1\}$, let's define $V_t^a(h)$ as the value function, $\sum_{k=t}^T \mathbb{E}^a(R_k|H_t = h)$, when a certain action is followed. It's evident from this that the ATE equals $T^{-1}\mathbb{E}[V_1^1(O_1) - V_1^0(O_1)]$. Let $\pi^b$ denote the behavior policy that generates the experimental data, i.e., $\pi_t^b(a|H_t) = \mathbb{P}(A_t = a|H_t)$ for any $a$ and $t$. Our objective lies in the design of an optimal behavior policy to generate high-quality data so that the mean squared error of the subsequent ATE estimator is minimized.

## 3 Design, Implementation and Evaluation in NMDPs

We focus on NDMPs in this section. We begin by proposing the optimal dynamic treatment allocation strategy to generate the experimental data. We next discuss some implementation details to implement the proposed design. Finally, we construct the ATE estimator from the data collected.

**Design**. Jiang and Li (2016) and Kallus and Uehara (2020) established the semiparametric efficiency bound, which is a nonparametric extension of the Cramer-Rao lower bound in parametric models (Bickel et al., 1993), for offline estimation of the average return under a general target policy in NMDPs. In essence, the efficiency bound corresponds to the smallest mean squared error (MSE) among a broad class of regular off-policy estimators. They further developed semiparametrically efficient estimators whose asymptotic MSEs reach this lower bound. Based on their results, one can easily show that the efficiency bound (EB) for the ATE equals:

$$\text{EB}_1(\pi^b) = T^{-2} \sum_{a \in \{0,1\}} \sum_{t=1}^T \mathbb{E}^{\pi^b}\left[\sigma_t(H_t, a) \prod_{k \leq t} \frac{\mathbb{I}(A_k = a)}{\pi_k^b(a|H_k)}\right]^2 + T^{-2}\text{Var}[V_1^1(O_1) - V_1^0(O_1)], \quad (1)$$

where $\sigma_t^2(H_k, a)$ denotes the conditional variance of the temporal difference error $R_t + V_{t+1}^a(H_{t+1}) - V_t^a(H_t)$ given $H_t$ and that $A_t = a$, and $\mathbb{I}(\bullet)$ denotes the indicator function. Note that the second term on the right-hand-side (RHS) is independent of the behavior policy. However, the first term is a function of $\pi^b$ in that: (i) the ratio in the first term explicitly involves $\pi^b$; (ii) the expectation $\mathbb{E}^{\pi^b}$ is taken with respect to the data generated by $\pi^b$.

The proposed design identifies the optimal $\pi^{b*}$ that minimizes (1). Before delving into the details of the proposed design, we first present the main idea behind it. A key observation is that the cumulative ratio $\prod_{k \leq t}[\mathbb{I}(A_k = a)/\pi_k^b(a|H_k)]$ appears on the RHS of (1) due to the use of sequential importance sampling (see e.g., Jiang and Li, 2016, Equation (4)) to account for the distributional shift between $\pi^b$ and the global policy that assigns action $a$ at each time. In general, the value of the cumulative ratio grows with the discrepancy between the target and behavior policy. Hence, it is plausible to

---

**Algorithm 1** Treatment allocation algorithm for NMDPs

---

**Input:** The burn-in period $m_0$ for each global policy and the termination day $n$.
1: Run each global policy for $m_0$ days and obtain $\{O_t^{(i)}, A_t^{(i)}, R_t^{(i)}\}_{i=1}^{2m_0}$.
2: **while** $2m_0 < m \le n$ **do**
3:      Using the collected data to estimate $V_1^a$ and $\sigma_*(\bullet, a)$ via (3). Obtain $\widehat{\sigma}_*(O_1^{(m)}, a)$.
4:      Assign $A_1^m$ according to $\widehat{\pi}_{1,m-1}^{b*}(a|O_1^{(m)}) = \widehat{\sigma}_*(O_1^{(m)}, a)/[\widehat{\sigma}_*(O_1^{(m)}, 1) + \widehat{\sigma}_*(O_1^{(m)}, 0)]$.
5:      Set $A_2^{(m)} = \cdots = A_T^{(m)} = A_1^{(m)}$.
6: **end while**
**Output:** $\{O_t^{(i)}, A_t^{(i)}, R_t^{(i)}\}_{i=1}^{n}$.

---

expect that an on-policy estimator, where all actions are determined according to the target policy, could potentially minimize the MSE.

We show in Theorem 1 below that the optimal design corresponds to a slight modification of the aforementioned on-policy design. Specifically, it randomly allocates the initial action with probabilities proportional to the standard deviations of the cumulative rewards (see Equation (2)), and assigns the same action afterwards.

**Theorem 1.** *In NMDPs, $\pi^{b*}$ that minimizes* (1) *satisfies: (i) for any $a \in \{0, 1\}$*

$$\pi_1^{b*}(a|O_1) = \frac{\sigma_*(O_1, a)}{\sigma_*(O_1, 0) + \sigma_*(O_1, 1)}, \tag{2}$$

*where $\sigma_*^2(O_1, a) = \mathbb{E}^a[\{\sum_t (R_t - \mathbb{E}^a R_t)\}^2 | O_1, A_1 = a]$; (ii) $\pi_2^{b*}(A_1|H_2) = \pi_3^{b*}(A_1|H_3) = \cdots = \pi_T^{b*}(A_1|H_T) = 1$ almost surely, or equivalently, $A_1 = A_2 = \cdots = A_T$ under $\pi^{b*}$.*

**Implementation**. It remains to estimate the standard deviation $\sigma_*$ to implement the proposed design. If historical data is available, we can use it to estimate $\sigma_*$. Otherwise, we can use data collected from the experiment to adaptively update this parameter. Initially, we run each global policy for $m_0$ days to generate the data. Next, on the $m$th day ($2m_0 < m \le n$), we estimate $\sigma_*$ using the experimental data collected so far. To this end, we begin by applying existing supervised learning algorithm to estimate the value function $V_1^a$ by regressing the cumulative rewards $\{\sum_t R_t^{(i)} : A_1^{(i)} = a, i < m\}$ on the initial observations $\{O_1^{(i)} : A_1^{(i)} = a, i < m\}$ where the superscript $i$ indicates that the data are collected on the $i$th day. Let $\widehat{V}_{1,m-1}^a$ denote the resulting estimator. We next employ supervised learning again to regress the squared residuals on the initial observations to estimate $\sigma_*(\bullet, a)$ as

$$\widehat{\sigma}_*(\bullet, a) = \arg\min_{\sigma(\cdot)} \sum_{i=1}^{m-1} \mathbb{I}(A_1^{(i)} = a)\left\{\left[\sum_t R_t^{(i)} - \widehat{V}_{1,m-1}^{(a)}(O_1^{(i)})\right]^2 - \sigma^2(O_1^{(i)})\right\}^2. \tag{3}$$

Finally, we plug $\widehat{\sigma}_*(O_1^{(m)}, a)$ into the RHS of (2) to obtain $\widehat{\pi}_{1,m-1}^{b*}$, assign $A_1^{(m)}$ according to $\widehat{\pi}_{1,m-1}^{b*}$ and set $A_2^{(m)} = \cdots = A_T^{(m)} = A_1^{(m)}$. We summarize our procedure in Algorithm 1.

**Evaluation**. Finally, we compute the ATE estimator using the experimental data. Several OPE algorithms, including value-based, importance sampling (IS), and doubly robust (DR) methods, are applicable for this purpose. DR estimators are known for achieving the efficiency bound (Kallus and Uehara, 2020). In our context, we suggest using the following online DR estimator for ATE,

$$\widehat{\text{ATE}}_1 = \sum_{a=0}^{1} \frac{(-1)^{a+1}}{T(n - 2m_0)} \sum_{i=2m_0+1}^{n} \left[\widehat{V}_{1,i-1}^a(O_1^{(i)}) + \frac{\mathbb{I}(A_1^{(i)} = a)}{\widehat{\pi}_{1,i-1}^{b*}(a|O_1^{(i)})}[\sum_t R_t^{(i)} - \widehat{V}_{1,i-1}^a(O_1^{(i)})]\right]. \tag{4}$$

Note that both the estimated value function $\widehat{V}_{1,i-1}^a$ and behavior policy $\widehat{\pi}_{1,i-1}^{b*}$ in (4) are computed during the data collection process. These nuisance functions are independent of the data $\{O_1^{(i)}, A_1^{(i)}, R_t^{(i)}\}$ used to construct the policy value. This cross-fitting procedure enables us to circumvent the need to impose certain metric entropy conditions (Díaz, 2020) and has been widely employed in the statistics and machine learning literature (Luedtke and Van Der Laan, 2016; Bibaut et al., 2021; Shi et al., 2021). By design, (4) can be calculated in an *online* manner, eliminating the need to store historical data. Moreover, we implement a burn-in procedure that discards the first $2m_0$

samples to construct $\widehat{\text{ATE}}$. This ensures the consistencies of $\widehat{V}^a_{1,i}$ and $\widehat{\pi}^{b*}_{1,i}$. Concurrently, value-based and IS-based methods are also suitable for estimating ATE. Provided that the nuisance functions are properly modeled, these estimators can achieve the efficiency bound as well (Uehara et al., 2020; Liao et al., 2021; Wang et al., 2021; Shi et al., 2023). The following theorem provides an upper bound for the MSE of the proposed ATE estimator.

**Theorem 2.** *Suppose that* $\min_i \widehat{\pi}^{b*}_{1,i} \geq \epsilon$ *and* $V^a_{1,i} \leq T R_{\max}$ *for some constants* $\epsilon > 0$ *and* $R_{\max} < \infty$, $\max_{a,i} \mathbb{E}|1/\widehat{\pi}^{b*}_{1,i}(a, O_1) - 1/\pi^b_1(a, O_1)|^2_2 \leq Ci^{-2\alpha_1}$ *and* $\max_{a,i} T^{-2}\mathbb{E}|\widehat{V}^a_{1,i}(O_1) - V^a_{1,i}(O_1)|^2_2 \leq Ci^{-2\alpha_2}$ *for some constants* $C < \infty$, $0 < \alpha_1, \alpha_2 < 1/2$. *Then we have*

$$\mathbb{E}(\widehat{\text{ATE}}_1 - ATE)^2 \leq \frac{EB_1(\pi^{b*})}{n - 2m_0} + O(\epsilon^{-1}C(n - 2m_0)^{-1-2\alpha_2}) + O(R^2_{\max}\sqrt{C}(n - 2m_0)^{-1-\alpha_1}).$$

In this case, the exponents $\alpha_1$ and $\alpha_2$ denote the convergence rates of the estimated behavior policy and value function, respectively. Note that the first term is the leading term, while the last two terms represent the estimation errors of the nuisance function estimators and converge at a rate much faster than $(n - 2m_0)^{-1}$. The error bound above is minimized when $m_0$ is zero. In such a scenario, the bound is asymptotically equal to $n^{-1}EB_1(\pi^{b*})$, which is equivalent to the MSE of an oracle ATE estimator. Specifically, the oracle estimator is a version of the DR estimator with correctly specified value function and behavior policy, and is constructed based on data collected over $n$ days from the optimal design $\pi^{b*}$. Thus, the proposed estimator is asymptotically optimal.

Next, we show how to construct the confidence interval of the ATE estimator. A key observation is that, under certain regularity conditions, the proposed estimator is asymptotically normal. More specifically, similar to Kallus and Uehara (2020), we have

$$\sqrt{n - 2m_0}(\widehat{\text{ATE}}_1 - \text{ATE}) \xrightarrow{d} N(0, EB_1(\pi^{b*})).$$

This motivates us to consider the following Wald-type confidence interval

$$[\widehat{\text{ATE}}_1 - \Phi^{-1}(1 - \alpha/2)\sqrt{EB_1(\pi^{b*})/(n - 2m_0)}, \widehat{\text{ATE}}_1 + \Phi^{-1}(1 - \alpha/2)\sqrt{EB_1(\pi^{b*})/(n - 2m_0)}],$$

where $\Phi^{-1}$ is the inverse cumulative distribution function of a standard normal random variable. It then suffices to estimate the asymptotic variance $EB_1(\pi^{b*})$ to construct asymptotically valid confidence intervals. Notice that $\widehat{\text{ATE}}_1$ can be represented as an average of martingale differences $\widehat{\text{ATE}}_1 = \sum_{i=2m_0+1}^{n} \psi^1_i/(n - 2m_0)$ where

$$\psi^1_i = \sum_{a=0}^{1} \frac{(-1)^{a+1}}{T}\left[\widehat{V}^a_{1,i-1}(O^{(i)}_1) + \frac{\mathbb{I}(A^{(i)}_1 = a)}{\widehat{\pi}^{b*}_{1,i-1}(a|O^{(i)}_1)}\left[\sum_t R^{(i)}_t - \widehat{V}^a_{1,i-1}(O^{(i)}_1)\right]\right].$$

We propose using the sample variance of $\{\psi^1_i\}_i$ to estimate $EB_1(\pi^{b*})$. Similar to Theorem 15 of Kallus and Uehara (2022), we can establish the consistency of the sampling variance estimator.

## 4 Design, Implementation and Evaluation in TMDPs and MDPs

We proceed to study the optimal design and the subsequent ATE estimation in TMDPs.

**Design.** According to Theorem 2 of Kallus and Uehara (2020), the efficiency bound for the ATE estimator equals

$$EB_2(\pi^b) = \frac{1}{T^2}\sum_{t=1}^{T}\sum_{a\in\{0,1\}} \mathbb{E}^{\pi^b}\left[\frac{\mathbb{I}(A_t = a)p^a_t(O_t)}{p^b_t(O_t, a)}\sigma_t(O_t, a)\right]^2 + \frac{1}{T^2}\text{Var}[V^1_1(O_1) - V^0_1(O_1)], \quad (5)$$

where $p^1_t(\bullet)$ ($p^0_t(\bullet)$) denotes the probability mass/density function of $O_t$ under the new policy (control), $p^b_t(\bullet, \bullet)$ denotes the probability mass/density function of the observation-action pair $(O_t, A_t)$ at time $t$ under the behavior policy, and $\sigma^2_t(O_t, a)$ is the conditional variance of the temporal difference error $R_t + V^a_{t+1}(O_{t+1}) - V^a_t(O_t)$ given the current observation $O_t$ and that $A_t = a$.

The efficiency bound in (5) bears a striking resemblance to that in (1). The sole difference lies in the replacement of the sequential IS ratio in (1) with the ratio of the marginal observation-action pair

under the Markov assumption. However, in contrast to NMDPs, the marginal distribution function $p_t^b$ in TMDPs cannot be represented in a closed-form as a function of $\pi^b$. This intricate dependence of $p_t^b$ on $\pi^b$ makes it exceptionally challenging to identify the optimal $\pi^{b*}$ that minimizes (5). To elaborate, let $\Pi^b$ represent the class of behavior policies that randomly assign the initial action, and set $\Pi^b = \{\pi^b : \pi_2^b(A_1|H_2) = \pi_3^b(A_1|H_3) = \cdots = \pi_T^b(A_1|H_T) = 1\}$. In Theorem 1, we show that the optimal behavior policy $\pi^{b*}$ belongs to $\Pi^b$ in NMDPs. However, this is not the case in TMDPs without additional assumptions.

**Proposition 1** (Informal Statement). *There exists a TMDP such that $\pi^{b*} \notin \Pi^b$.*

Identifying the exact optimal design remains challenging. Motivated by the simplicity of the policy class $\Pi^b$, we shift our focus to finding the optimal *in-class* behavior policy within $\Pi^b$ that minimizes (5). We restrict the treatment assignment policies to $\Pi^b$ for two primary reasons. First, the optimal design for NMDP resides in class $\Pi^b$, leveraging this result allows us to anticipate that in-class behavior policies from $\Pi^b$ will perform well in TMDPs/MDPs, which are subclasses of NMDP; Second, frequently alternating treatments can introduce significant carryover bias in policy evaluation (Hu and Wager, 2022). By focusing on $\Pi^b$, we avoid distributional shifts between the behavior policy and the target policy, effectively mitigating carryover bias. To further simplify the analysis, we consider scenarios where the number of decision stages $T$ approaches infinity. The following theorem provides the form of an optimal $\pi^{b*} \in \Pi^b$ that *asymptotically* minimizes (5) under a mild $\beta$-mixing condition. Additionally, it shows that $\pi^{b*}$ is optimal among all history-dependent policies under certain constancy conditions.

**Theorem 3.** *Suppose the $\beta$-mixing condition holds such that $\lim_{t \to \infty} \mathbb{E} \sup_{a,o} |p_t^a(o|O_1) - p_t^a(o)| \to 0$ where $p_t^a(\bullet|O_1)$ denotes the probability mass function of $O_t$ given $O_1$ following the action $a$. Then an asymptotically optimal in-class behavior policy $\pi^{b*}$ satisfies (i) for any $a \in \{0, 1\}$,*

$$\pi_1^{b*}(a|O_1) = \pi_1^{b*}(a) = \frac{\sigma_{a*}}{\sigma_{1*} + \sigma_{0*}}, \text{ where } \sigma_{a*}^2 = \sum_{t=1}^T \mathbb{E}^a[\sigma_t^2(O_t, 1)]; \tag{6}$$

*(ii) $\pi_3^{b*}(A_1|H_2) = \pi_3^{b*}(A_1|H_3) = \cdots = \pi_T^{b*}(A_1|H_T) = 1$ almost surely. In other words, for $\pi^{b*}$ that satisfies (i) and (ii), we have $\lim_T T[EB_2(\pi^{b*}) - EB_2(\pi^b)] \le 0$ for any $\pi^b \in \Pi^b$. Additionally, suppose both $\sigma_t^2(o, 1)$ and $\sigma_t^2(o, 0)$ are constant as functions of $o$ and $t$. Then $\lim_T T[EB_2(\pi^{b*}) - EB_2(\pi^b)] \le 0$ for any $\pi^b$.*

The $\beta$-mixing condition essentially requires the Markov chain to be ergodic. Similar conditions have been imposed in the literature (Bhandari et al., 2018; Zou et al., 2019; Luckett et al., 2020; Kallus and Uehara, 2022; Li et al., 2022; Shi et al., 2022, 2023). It is also weaker than the independence assumption that requires the transition tuples to be independent over time and has been frequently imposed in the literature. The constancy condition is automatically satisfied when the temporal difference error is independent of the current observation and the time step. Different from the optimal design in NMDPs, the initial policy in (6) is independent of the initial observation, thanks to the mixing condition.

**Implementation**. We summarize the procedure in Algorithm 2.

**Evaluation**. Similar to Section 3, we can apply value-based, IS-based, or doubly robust estimator to learn the ATE from the collected data. Consider the double reinforcement learning estimator (DRL, Kallus and Uehara, 2020) as an example. A key observation is that the marginal observation-action probability distribution function $p_t^b(O_t, A_t)$ under $\pi^{b*}$ equals $\pi_1^*(A_t)p_t^{A_t}(O_t)$. As such, the resulting marginal ratio $\mathbb{I}(A_t = a)p_t^a(O_t)/p_t^b(O_t, a)$ is equal to $\mathbb{I}(A_t = a)/\pi_1^*(A_t)$, or equivalently $\mathbb{I}(A_1 = a)/\pi_1^*(A_1)$, independent of $O_t$. Consequently, one can show the resulting DRL is reduced to

$$\widehat{\text{ATE}}_2 = \sum_{a=0}^1 \frac{(-1)^{a+1}}{T(n - 2m_0)} \sum_{i=2m_0+1}^n \left\{ \widehat{V}_{1,i-1}^a(O_1^{(i)}) + \frac{\mathbb{I}(A_1^{(i)} = a)}{\widehat{\pi}_{1,i-1}^{b*}(a)} [\sum_t R_t^{(i)} - \widehat{V}_{1,i-1}^a(O_1^{(i)})] \right\},$$

where $\widehat{V}_{1,i}^a$ and $\widehat{\pi}_1^{b*}(a)$ denotes the estimated value function and estimated initial allocation probability using data from the first $i$th days; see Step 3 of Algorithm 2 for the estimation procedure. Similar to $\widehat{\text{ATE}}_1$ in (4), $\widehat{\text{ATE}}_2$ can be updated in an online manner without storing historical data. We provide an upper bound for the MSE of $\widehat{\text{ATE}}_2$ below.

---

**Algorithm 2** Treatment allocation algorithm for TMDPs

---

**Input:** The burn-in period $m_0$ for each global policy and the termination day $n$.

1: Run each global policy for $m_0$ days and obtain $\{O_t^{(i)}, A_t^{(i)}, R_t^{(i)}\}_{i=1}^{2m_0}$.
2: **while** $2m_0 < m \leq n$ **do**
3:  Obtain $\widehat{V}_{t,m-1}^a$ by regressing $\{\sum_{j=t}^T R_t^{(i)} : i < m, A_1^{(i)} = a\}$ on $\{O_t^{(i)} : i < m, A_1^{(a)} = a\}$.
4:  For $a \in \{0, 1\}$, set

$$\widehat{\sigma}_{a*}^2 = \frac{\sum_{i<m}\sum_{t=1}^T [R_t^{(i)} + \widehat{V}_{t+1,i}^a(O_{t+1}^{(i)}) - \widehat{V}_{t,i}^a(O_t^{(i)})]^2 \mathbb{I}(A_1^{(i)} = a)}{\sum_{i<m}\mathbb{I}(A_1^{(i)} = a)}.$$

5:  Assign $A_1^{(m)}$ according to $\widehat{\pi}_{1,m-1}^{b*}(a) = \widehat{\sigma}_{a*}/(\widehat{\sigma}_{1*} + \widehat{\sigma}_{0*})$.
6:  Set $A_2^{(m)} = \cdots = A_T^{(m)} = A_1^{(m)}$.
7: **end while**
**Output:** $\{O_t^{(i)}, A_t^{(i)}, R_t^{(i)}\}_{i=1}^n$.

---

**Theorem 4.** *Suppose that* $\min_i \widehat{\pi}_{1,i}^{b*} \geq \epsilon$ *and* $V_{1,i}^a \leq TR_{\max}$ *for some constants* $\epsilon > 0$ *and* $R_{\max} < \infty$, $\max_{a,i}\mathbb{E}|1/\widehat{\pi}_{1,i}^{b*}(a) - 1/\pi_1^{b*}(a)|_2^2 \leq Ci^{-2\alpha_1}$ *and* $\max_{a,i} T^{-2}\mathbb{E}|\widehat{V}_{1,i}^a(O_1) - V_{1,i}^a(O_1)|_2^2 \leq Ci^{-2\alpha_2}$ *for some constants* $C < \infty$, $0 < \alpha_1, \alpha_2 < 1/2$. *Then we have*

$$\mathbb{E}(\widehat{ATE}_2 - ATE)^2 \leq \frac{EB_2(\pi^{b*})}{n - 2m_0} + O(C\epsilon^{-1}(n - 2m_0)^{-1-2\alpha_2}) + O(\sqrt{C}R_{\max}^2(n - m_0)^{-1-\alpha_1}).$$

Similar to Theorem 2, the MSE of the proposed ATE estimator relies on $m_0$, the estimated behavior policy and value function. The first term is asymptotically equal to $n^{-1}EB_2(\pi^{b*})$ when $m_0$ is much smaller than $n$. This suggests that the proposed estimator is asymptotically optimal in TDMPs.

Analogous to the procedures in Section 3, we can similarly establish the asymptotic normality of $\widehat{ATE}_2$, i.e., $\sqrt{n - 2m_0}(\widehat{ATE}_2 - ATE) \overset{d}{\to} N(0, EB_2(\pi^{b*}))$. The corresponding $1 - \alpha$ confidence interval can be constructed by

$$[\widehat{ATE}_2 - \Phi^{-1}(1 - \alpha/2)\sqrt{EB_2(\pi^{b*})/(n - 2m_0)}, \widehat{ATE}_2 + \Phi^{-1}(1 - \alpha/2)\sqrt{EB_2(\pi^{b*})/(n - 2m_0)}],$$

where the unknown asymptotic variance $EB_2(\pi^{b*})$ can be similarly estimated via the sampling variance estimator.

**MDPs.** Finally, we consider MDPs. To save space, we briefly introduce our proposal here and relegate the technical details to the supplementary article. The proposed design is built upon the efficiency bound developed by Liao et al. (2022) in the average reward infinite horizon setting. To simplify the analysis, similar to our proposal in TMDPs, we focus on in-class optimal designs and consider an asymptotic regime where $T \to \infty$. The following theorem summarizes the properties of the proposed behavior policy $\tilde{\pi}^{b*} \in \Pi^b$.

**Theorem 5** (Information Statement). *Suppose the $\beta$-mixing condition holds. Then $\tilde{\pi}^{b*}$ asymptotically minimizes the efficiency bound among all $\pi^b \in \Pi^b$. In addition, under a constancy condition, it is asymptotically optimal among all policies.*

To learn the ATE, we construct an online doubly robust estimator in the supplementary article based on the estimated relative value functions (Puterman, 2014) and the designed behavior policy.

## 5 Experiment

This section presents four experiments designed to evaluate different designs. The environments include a tabular example with binary observation variables, an example with continuous observations, a small-scale synthetic dispatch example, and a city-scale real data-based dispatch simulator. We investigate the proposed treatment allocation strategies designed for NMDPs and MDPs. We also implement the following three allocation designs for comparison:

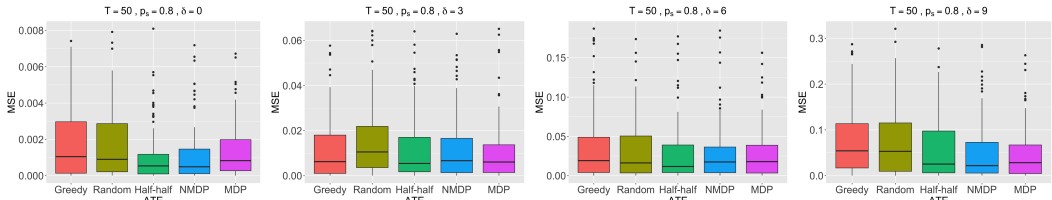

Figure 2: Boxplots of the MSEs of the allocation methods for $p_s = 0.8$ and $T = 50$ in Example 5.1: the four panels correspond to $\delta = 0, 3, 6$, and 9, respectively.

  (i) Random: This uniform random treatment allocation design assumes $\mathbb{P}(A_{i,t} = 1) = 1/2$. All $A_{i,t}$s are independent.

 (ii) Half-half: This design applies the global treatment for the first $n/2$ days and the global control for the remaining days.

(iii) Greedy: This design uses the $\epsilon$-greedy algorithms that select the current-best treatment, which maximizes the Q-function, with probability $1 - \epsilon$, and employs a uniform random policy with probability $\epsilon$.

We note that (iii) is widely used in the RL literature for online regret minimization. We compare the mean squared errors (MSEs) of the estimates of the average treatment effect based on 200 replicates, and present the results subsequently. We set the burn-in period $m_0$ to $n/4$ in all the experiments. Detailed information on the data-generating processes can be found in Section S2 of the supplementary material.

**Example 5.1** (**Binary Observations**). In this example, we consider binary observation variables. To better understand the results, we introduce some hyper-parameters that describe the system dynamics. Let $p_s$ denote the marginal probability that the future observation $O_{t+1}$ equals 1 if $A_t = 1$, and it is $1 - p_s$ if $A_t = 0$. Additionally, we use $\delta \in \{0, 3, 6, 9\}$ to characterize the difference of the conditional variance of the reward between the treatment and the control. A larger value of $\delta$ indicates a greater difference.

Figure 2 presents boxplots of the MSEs for the allocation methods when $p_s = 0.8$, $T = 50$, and $n = 50$. It is evident that the proposed treatment allocation methods generally outperform the alternatives, offering lower median MSE and smaller variability in most scenarios. However, when $\delta = 0$, the half-half design outperforms ours. This outcome is expected since in this case, the optimal allocation rule assigns the initial action with equal probability, i.e., $\pi_1^{b*} = 0.5$. As $\delta$ increases, our proposed estimator surpasses the others in terms of achieving a smaller MSE. Table 1 in the supplementary material presents the means and standard deviations of the MSEs for $T \in \{10, 30, 50\}$ and $p_s \in \{0.5, 0.8\}$. In these scenarios, our proposed designs consistently outperform the alternatives.

**Example 5.2** (**Continuous Observations**). In this example, we use $\delta_s$ to denote the difference in the conditional variance of the observation variable between the treatment and the control, and $\delta$ to describe the difference of the conditional variance of the reward.

Figure 3 presents the boxplots of all MSEs for $\delta_s = 1$ and $T = 50$ with $n = 50$. Table 2 of the supplementary material includes the Monte Carlo averages and standard errors of MSEs for $T \in \{10, 30, 50\}$ and $\delta_s \in \{0, 1\}$. These results are comparable to those in Example 5.1, demonstrating the superior performance of the proposed design when $\delta$ is moderately large. This is expected, as the proposed method takes into account the conditional variance of the temporal difference error.

**Example 5.3** (**Synthetic Dispatch**). Following Xu et al. (2018), we construct a small-scale synthetic dispatch environment to estimate the treatment effect of different order dispatch policies. Specifically, we simulate drivers and orders in a $9 \times 9$ spatial grid with a duration of 20 time steps per day. Orders will be canceled if not being responded for a long time. We compare the MDP order dispatch strategy (Xu et al., 2018) against the distance-based dispatch method which minimizes the total distance between drivers and passengers. The reward of interest is given by the total revenue. The observation variables are set to the number of orders and the number of drivers at each time. The number of days is set to be $n \in \{30, 50, 100\}$. All the methods are tested using 100 orders with the number of drivers being either generated from the uniform distribution $U(25, 30)$, or being fixed to 25, 50. A detailed description of the environment can be found in the supplementary document.

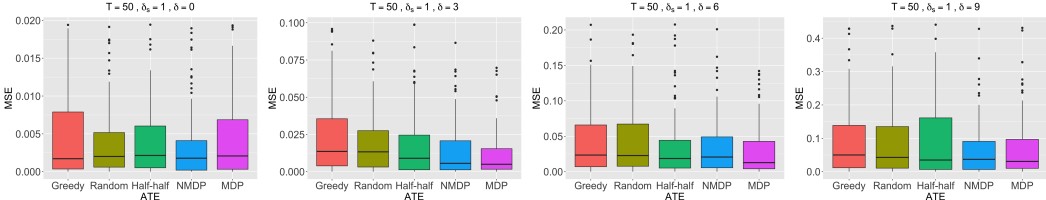

Figure 3: Boxplots of the MSEs of the allocation methods corresponding to $\delta_s = 1$ and $T = 50$ in Example 5.2: the four panels correspond to $\delta = 0, 3, 6, 9$, respectively.

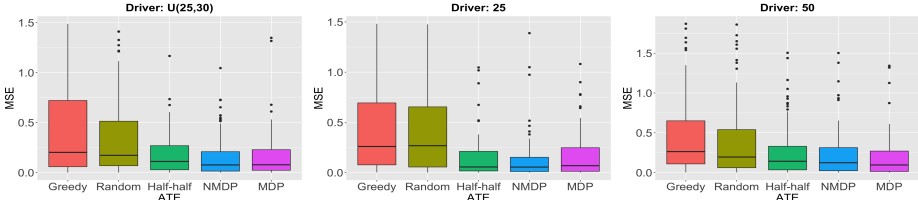

Figure 4: Boxplots of the MSEs of the allocation methods in Example 5.3 with $n = 50$: the three panels correspond to drivers generated from $U(25, 30)$ or fixed as 25 and 50, respectively.

Figure 4 presents the boxplots of MSEs for $n = 50$ with different numbers of drivers. Table 3 of the supplementary material reports the values of MSEs of the four methods. As expected, the MSEs of all the methods decrease as the increase of the number of days $n$. The proposed method achieves the best performance in almost all scenarios.

**Example 5.4** (**Real-Data Based Dispatch**). We evaluate the proposed treatment allocation method on a dispatch simulator based on a city-scale order-driver historical dataset from a world-leading ride-sharing platform. The dataset consists of temporal spatial information of orders or drivers and numerical features of them. We adopt the simulator in Tang et al. (2019) to generate data based on the historical dataset. The distributions of drivers and orders are set to be identical to the distributions of historical data. Similar to Example 5.3, the two policies being compared are the MDP strategy and the distance-based dispatch method. The reward of interest is the total Gross Merchandise Volume (GMV), and the observation variables include the number of drivers and the number of orders.

Since the ground truth of the average treatment effect is large, we calculate the relative MSE of the estimated average treatment effect for each method. RMSEs of ATE estimates for distinct allocation designs are reported in Figure 5 with the number of days $n = 7$ and $n = 10$. It can be seen that the proposed method outperforms all its counterparts. Meanwhile, as the episode $n$ gets larger, the RMSEs of most allocation designs become smaller.

## Acknowledgement

Li's research is partially supported by the National Science Foundation of China 12101388, CCF-DiDi GAIA Collaborative Research Funds for Young Scholars and Program for Innovative Research Team of Shanghai University of Finance and Economics. Shi's research is partially supported by an EPSRC grant EP/W014971/1. Zhou's work is partially supported by National Natural Science Foundation of China 12001356, "Chenguang Program" supported by Shanghai Education Development

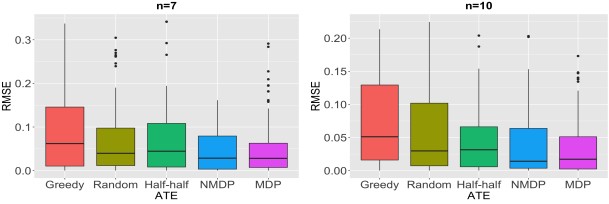

Figure 5: Boxplots of the RMSEs of the allocation methods in Example 5.4 with $n = 7$ and $n = 10$.

Foundation and Shanghai Municipal Education Commission, Open Research Projects of Zhejiang Lab NO.2022RC0AB06, Shanghai Research Center for Data Science and Decision Technology. We thank the anonymous referees and the meta reviewer for their constructive comments, which have led to a significant improvement of the earlier version of this article.

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
