# Supplementary Material for Optimal Treatment Allocation for Efficient Policy Evaluation in Sequential Decision Making

**Ting Li**[1]  **Chengchun Shi**[2]  **Jianing Wang**[1]  **Fan Zhou**[1]  **Hongtu Zhu**[3] *

[1]School of Statistics and Management, Shanghai University of Finance and Economics
[2]Department of Statistics, London School of Economics and Political Science
[3]Department of Biostatistics, University of North Carolina at Chapel Hill

tingli@mail.shufe.edu.cn, C.Shi7@lse.ac.uk, jianing.wang@163.sufe.edu.cn
zhoufan@mail.shufe.edu.cn,htzhu@email.unc.edu

This supplementary material contains design, implementation and evaluation in MDPs, detailed data-generating processes of the experiment, additional experiment results and all the proofs.

## S1  Design, Implementation and Evaluation in MDPs

In this section, we study the optimal design and the subsequent ATE estimation in time-homogeneous MDPs. We consider an infinte horizon setting where the ATE equals

$$\text{ATE} = \underbrace{\lim_{H \to \infty} \frac{1}{H} \sum_{t=1}^{H} \mathbb{E}^1(R_t)}_{J(1)} - \underbrace{\lim_{H \to \infty} \frac{1}{H} \sum_{t=1}^{H} \mathbb{E}^0(R_t)}_{J(0)}.$$

**Design**. Similar to Kallus and Uehara (2022) and Liao et al. (2022), the efficiency bound for the above ATE estimator is given by

$$\text{EB}_3(\pi^b) = \frac{1}{T^2} \sum_{t=1}^{T} \sum_{a \in \{0,1\}} \mathbb{E}^{\pi^b} \Big[ \frac{\mathbb{I}(A_t = a) p^a(O_t)}{\sum_{j=1}^{T} p_j^b(O_t, a)/T} \sigma(O_t, a) \Big]^2, \tag{S1}$$

where $p^1(\bullet)$ $(p^0(\bullet))$ denotes the stationary probability mass/density function of $O_t$ under the new policy (control), $p_j^b$ denotes the visitation probability of the observation-action pair $(O_t, A_t)$ at time $j$ under the behavior policy, and $\sigma^2(O_t, a)$ is the conditional variance of the temporal difference error $R_t + V^a(O_{t+1}) - V^a(O_t) - J(a)$ given the current observation $O_t$ and that $A_t = a$ where $V_a$ denotes the relative value function $\sum_{t=1}^{\infty} \mathbb{E}^1[R_t - J(1)|O_1]$ (Puterman, 2014; Liao et al., 2022).

Compared to those under TMDP in (5), the probability function $p^a(O_t)$ and the conditional variance $\sigma^2(O_t, a)$ in the efficiency bound in (S1) do not vary across time, due to time stationarity. Similar to TMDPs, it is challenging to find the optimal $\pi^{b*}$ that minimizes (S1) because of the convoluted dependency of $p_t^b$ on $\pi^b$. To address this, we also restrict our attention to an optimal *in-class* behavior policy belonging to $\Pi^b = \{\pi^b : \pi_2^b(A_1|H_2) = \pi_3^b(A_1|H_3) = \cdots = \pi_T^b(A_1|H_T) = 1\}$.

We derive the form of an asymptotically optimal $\pi^{b*}$ within the class $\Pi^b$ under the $\beta$-mixing condition, and establishes its optimality among all policies under certain constancy conditions. We use $\lambda$ to denote the counting or Lebesgue measure, depending on whether the variable of interest is discrete or continuous.

---

*The first two authors contribute equally to this paper. Address for correspondence: Hongtu Zhu, Ph.D., E-mail: htzhu@email.unc.edu.

37th Conference on Neural Information Processing Systems (NeurIPS 2023).

**Algorithm S1** Treatment allocation algorithm for MDPs
___
**Input:** The burn-in period $m_0$ for each global policy and the termination time $n$.

1: Run each global policy for $m_0$ days and obtain $\{O_t^{(i)}, A_t^{(i)}, R_t^{(i)}\}_{i=1}^{2m_0}$.
2: **while** $2m_0 < m \le n$ **do**
3:     Obtain $\widehat{V}_i^a$ and $\widehat{J}_i(a)$ by solving the Bellman equation

$$\mathbb{E}^a[R_t^{(i)} + V^a(O_{t+1}^{(i)}) - J(a)|O_t^{(i)}] = V^a(O_t^{(i)}),$$

    using the data subset $\{(O_t^{(i)}, R_t^{(i)}) : i < m, 1 \le t \le T, A_1^{(i)} = a\}$.
4:     For $a \in \{0, 1\}$, set

$$\widehat{\sigma}_{a*}^2 = \frac{\sum_{i<m}\sum_{t=1}^T [R_t^{(i)} + \widehat{V}_i^a(O_{t+1}^{(i)}) - \widehat{V}_i^a(O_t^{(i)}) - \widehat{J}_i(a)]^2 \mathbb{I}(A_1^{(i)} = a)}{\sum_{i<m}\mathbb{I}(A_1^{(i)} = a)}.$$

5:     Assign $A_1^{(m)}$ according to $\widehat{\pi}_{1,m-1}^{b*}(a) = \widehat{\sigma}_{1*}/(\widehat{\sigma}_{1*} + \widehat{\sigma}_{0*})$.
6:     Set $A_2^{(m)} = \cdots = A_T^{(m)} = A_1^{(m)}$.
7: **end while**
**Output:** $\{O_t^{(i)}, A_t^{(i)}, R_t^{(i)}\}_{i=1}^n$.
___

**Theorem S1.** *Suppose the $\beta$-mixing condition holds such that $\lim_{t\to\infty} \mathbb{E}\sup_{a,o_t} |p^a(o_t|O_1) - p^a(o_t)| \to 0$ where $p^a(\bullet|O_1)$ denotes the probability mass function given $O_1$ following the action $a$. Then an asymptotically optimal in-class behavior policy $\pi^{b*}$ satisfies (i) for any $a \in \{0,1\}$,*

$$\pi_1^{b*}(a|O_1) = \frac{\sigma_{a*}}{\sigma_{1*} + \sigma_{0*}} \text{ where } \sigma_{a*}^2 = \int_o \sigma^2(o,a)p^a(o)d\lambda(o); \tag{S2}$$

*(ii) $\pi_2^{b*}(A_1|H_2) = \pi_3^{b*}(A_1|H_3) = \cdots = \pi_T^{b*}(A_1|H_T) = 1$ almost surely. In other words, for $\pi^{b*}$ that satisfies (i) and (ii), we have $\lim_T T[EB_3(\pi^{b*}) - EB_3(\pi^b)] \le 0$ for any $\pi^b \in \Pi^b$. Additionally, suppose both $\sigma^2(o,1)$ and $\sigma^2(o,0)$ are constant as functions of o. Then $\lim_T T[EB_3(\pi^{b*}) - EB_3(\pi^b)] \le 0$ for any $\pi^b$.*

The optimal policy derived in Theorem S1 has a similar form to the optimal policy for TMDPs. The initial policy in (S2) also depends on the conditional variance of the temporal difference error. However, the conditional variance function does not vary across the time $t$.

**Implementation**. We summarize the procedure for MDPs in Algorithm S1, which is similar to the procedure for TMDPs.

**Evaluation**. Similar to TMDP, we learn the ATE from the collected data by using the following online estimator,

$$\widehat{ATE}_3 = \sum_{a=0}^1 \frac{(-1)^{a+1}}{T(n-2m_0)} \sum_{i=2m_0+1}^n \left[\widehat{V}_{i-1}^a(O_1^{(i)}) + \frac{\mathbb{I}(A_1^{(i)} = a)}{\widehat{\pi}_{1,i-1}^{b*}(a)}[\sum_t R_t^{(i)} - \widehat{V}_{i-1}^a(O_1^{(i)})]\right],$$

where $\widehat{V}_i^a$ denotes the estimated value function using data from the first $i$th days. The proposed ATE estimator also takes advantage of the fact that the marginal observation-action probability distribution function $T^{-1}\sum_{j=1}^T p_j^b(O_t, A_t)$ is asymptotically equivalent to $\pi_1^*(A_t)p^{A_t}(O_t)$. Hence, the resulting marginal ratio $\mathbb{I}(A_t = a)p^a(O_t)/T^{-1}\sum_j p_j^b(O_t, a)$ can be replaced by $\mathbb{I}(A_t = a)/\pi_1^*(A_t)$, or equivalently $\mathbb{I}(A_1 = a)/\pi_1^*(A_1)$, independent of $O_t$. Similar to $\widehat{ATE}_1$ and $\widehat{ATE}_2$, $\widehat{ATE}_3$ can be updated in an online manner without storing historical data.

## S2 Additional Experiment Results

In this section, we present details of the data-generating process for Section 5 and additional experiment results.

The proposed treatment allocation methods and the $\epsilon$-greedy method rely on the estimation of value functions. For our method, we assign the treatment to the first $n/4$ samples, the control to the

subsequent $n/4$ samples, and then sequentially assign the treatment using the $\hat{\pi}^{b*}$ estimated from the data collected so far. For the $\epsilon$-greedy method, we choose $\epsilon = 0.05$. and randomly assign the treatment to the first $n/2$ samples, and then sequentially assign the treatment using the Q-function from the data collected so far.

**Example 5.1 (Continued).** In this example, we generate binary state variables as follows,

$$\mathbb{P}(O_{i,t+1} = 1) = p_s\mathbb{I}(A_{i,t} = 1) + (1 - p_s)\mathbb{I}(A_{i,t} = 0).$$

Then we generate the reward function via

$$R_{i,t+1} = 10 + 2A_{i,t} + 0.25O_{i,t}A_{i,t} \times (0.04 + 0.02O_{i,t}) + (1 + \delta A_{i,t})\xi_{i,t},$$

where $\xi_{i,t}$ is generated from the standard normal distribution $N(0, 1)$. We consider $n = 50$, $T \in \{10, 30, 50\}$, $p_s \in \{0.5, 0.8\}$, $\delta \in \{0, 3, 6, 9\}$ and calculate the mean squared errors (MSEs) of the estimates of the average treatment effect based on 200 replicates. The true average treatment effect is calculated by the difference in the empirical average of rewards between the treatment and the control groups using a sample of 1000 trajectories.

Table 1 gives the means and standard deviations of the MSEs. Figure S1 presents additional boxplots of the MSEs for the allocation methods when $p_s = 0.8$, $T \in \{10, 30\}$. The results show that when $\delta = 0$, the proposed methods behave similarly to the competing methods. However, when the conditional variance of the reward between the treatment and the control becomes large ($\delta$ becomes large), the proposed methods outperform other methods.

Table 1: Simulation results for Example 5.1 of Monte Carlo averages with standard errors in parentheses of MSEs for the average treatment effect based on 200 replicates.

| $T$ | $p_s$ | $\delta$ | Greedy | Random | Half-half | NMDP | TMDP |
|---|---|---|---|---|---|---|---|
| 10 | 0.5 | 0 | 0.008(0.010) | 0.009(0.010) | 0.008(0.012) | 0.008(0.013) | 0.009(0.011) |
| | 0.5 | 3 | 0.091(0.165) | 0.077(0.105) | 0.086(0.099) | 0.043(0.088) | 0.049(0.059) |
| | 0.5 | 6 | 0.210(0.297) | 0.239(0.374) | 0.184(0.245) | 0.130(0.173) | 0.102(0.133) |
| | 0.5 | 9 | 0.558(0.757) | 0.482(0.734) | 0.443(0.562) | 0.240(0.387) | 0.280(0.428) |
| | 0.8 | 0 | 0.011(0.015) | 0.009(0.013) | 0.009(0.011) | 0.008(0.010) | 0.010(0.012) |
| | 0.8 | 3 | 0.119(0.302) | 0.092(0.111) | 0.069(0.090) | 0.055(0.071) | 0.052(0.075) |
| | 0.8 | 6 | 0.341(0.559) | 0.329(0.399) | 0.220(0.296) | 0.137(0.164) | 0.140(0.186) |
| | 0.8 | 9 | 1.002(1.481) | 0.380(0.528) | 0.414(0.595) | 0.267(0.379) | 0.216(0.359) |
| 30 | 0.5 | 0 | 0.003(0.005) | 0.003(0.004) | 0.003(0.004) | 0.002(0.005) | 0.003(0.003) |
| | 0.5 | 3 | 0.031(0.054) | 0.018(0.029) | 0.017(0.025) | 0.018(0.025) | 0.017(0.024) |
| | 0.5 | 6 | 0.092(0.177) | 0.072(0.101) | 0.057(0.071) | 0.058(0.074) | 0.052(0.079) |
| | 0.5 | 9 | 0.139(0.229) | 0.158(0.226) | 0.156(0.229) | 0.094(0.131) | 0.102(0.137) |
| | 0.8 | 0 | 0.004(0.006) | 0.004(0.005) | 0.003(0.004) | 0.004(0.004) | 0.003(0.004) |
| | 0.8 | 3 | 0.029(0.042) | 0.034(0.047) | 0.026(0.033) | 0.018(0.030) | 0.015(0.024) |
| | 0.8 | 6 | 0.077(0.110) | 0.097(0.122) | 0.066(0.078) | 0.039(0.055) | 0.049(0.069) |
| | 0.8 | 9 | 0.295(0.457) | 0.189(0.236) | 0.146(0.200) | 0.093(0.153) | 0.115(0.163) |
| 50 | 0.5 | 0 | 0.002(0.003) | 0.002(0.002) | 0.001(0.002) | 0.002(0.002) | 0.002(0.002) |
| | 0.5 | 3 | 0.020(0.032) | 0.012(0.016) | 0.015(0.026) | 0.015(0.019) | 0.012(0.015) |
| | 0.5 | 6 | 0.062(0.091) | 0.050(0.076) | 0.042(0.069) | 0.031(0.050) | 0.026(0.041) |
| | 0.5 | 9 | 0.110(0.163) | 0.087(0.119) | 0.106(0.147) | 0.060(0.086) | 0.066(0.093) |
| | 0.8 | 0 | 0.003(0.005) | 0.002(0.003) | 0.001(0.002) | 0.001(0.002) | 0.002(0.002) |
| | 0.8 | 3 | 0.029(0.047) | 0.019(0.024) | 0.014(0.019) | 0.013(0.016) | 0.012(0.017) |
| | 0.8 | 6 | 0.088(0.137) | 0.044(0.062) | 0.034(0.052) | 0.029(0.037) | 0.030(0.036) |
| | 0.8 | 9 | 0.157(0.280) | 0.106(0.138) | 0.068(0.087) | 0.057(0.074) | 0.054(0.074) |

**Example 5.2 (Continued).** In this example, we consider three continuous observation variables and generate the transition of observation variables and reward functions via

$$
\begin{aligned}
O_{i,t+1,1} &= 0.5O_{i,t,1} + 2\epsilon_{i,t,1}, \\
O_{i,t+1,2} &= 0.25O_{i,t,2} + 0.2A_{i,t} + (2 + \delta_s)\epsilon_{i,t,1}, \\
O_{i,t+1,3} &= 0.6O_{i,t,3} + 0.05O_{i,t,3}A_{i,t} + 0.5A_{i,t} + \epsilon_{t,1} \\
R_{i,t+1} &= 10 + 2A_{i,t} - 0.4O_{i,t,3} + 0.2O_{i,t,2} + 0.25O_{i,t,1}A_{i,t} \times (0.04 + 0.02O_{i,t,1}) + (1 + \delta A_{i,t})\xi_{i,t},
\end{aligned}
$$

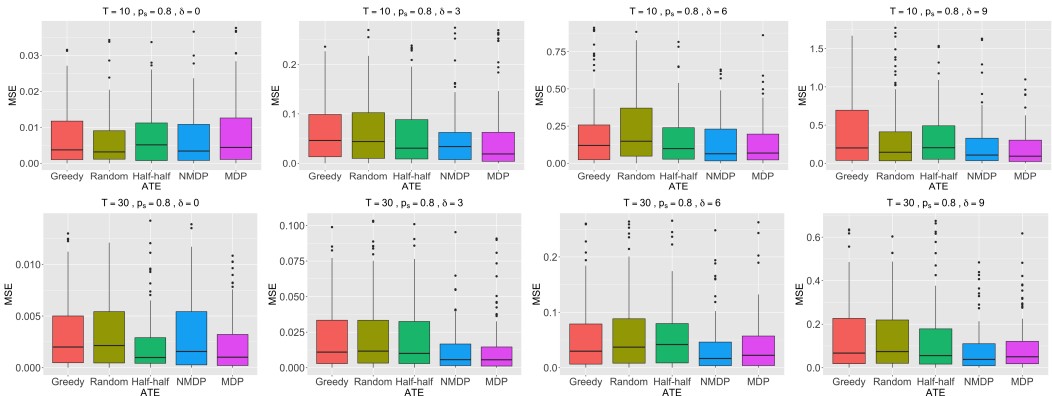

Figure S1: Boxplots of the MSEs of the allocation methods for $p_s = 0.8$ and $T \in \{10, 30\}$ in Example 5.1: the four panels correspond to $\delta = 0, 3, 6$, and $9$, respectively.

Table 2: Simulation results for Example 5.2 of Monte Carlo averages with standard errors in parentheses of MSEs for the average treatment effect based on 200 replicates.

| $T$ | $\delta_s$ | $\delta$ | Greedy | Random | Half-half | NMDP | TMDP |
|---|---|---|---|---|---|---|---|
| 10 | 0 | 0 | 0.227(0.678) | 0.064(0.161) | 0.040(0.101) | 0.023(0.031) | 0.022(0.031) |
| | | 3 | 0.289(0.581) | 0.180(0.501) | 0.125(0.182) | 0.088(0.170) | 0.078(0.118) |
| | | 6 | 0.612(1.035) | 0.451(0.821) | 0.329(0.622) | 0.188(0.226) | 0.225(0.294) |
| | | 9 | 0.820(1.219) | 0.822(0.982) | 0.556(0.792) | 0.306(0.423) | 0.348(0.513) |
| | 1 | 0 | 0.117(0.464) | 0.101(0.242) | 0.050(0.152) | 0.026(0.037) | 0.032(0.078) |
| | | 3 | 0.368(0.802) | 0.172(0.365) | 0.146(0.263) | 0.080(0.118) | 0.088(0.136) |
| | | 6 | 0.536(0.904) | 0.465(0.682) | 0.253(0.357) | 0.209(0.349) | 0.180(0.319) |
| | | 9 | 0.846(1.148) | 0.832(1.023) | 0.553(0.804) | 0.342(0.413) | 0.412(0.499) |
| 30 | 0 | 0 | 0.015(0.022) | 0.013(0.037) | 0.006(0.010) | 0.006(0.008) | 0.006(0.008) |
| | | 3 | 0.062(0.096) | 0.045(0.059) | 0.028(0.042) | 0.027(0.036) | 0.018(0.021) |
| | | 6 | 0.190(0.373) | 0.087(0.099) | 0.066(0.101) | 0.045(0.062) | 0.078(0.118) |
| | | 9 | 0.280(0.481) | 0.207(0.306) | 0.137(0.206) | 0.108(0.134) | 0.093(0.110) |
| | 1 | 0 | 0.038(0.140) | 0.024(0.117) | 0.008(0.011) | 0.009(0.012) | 0.007(0.008) |
| | | 3 | 0.057(0.088) | 0.050(0.079) | 0.022(0.039) | 0.022(0.030) | 0.026(0.041) |
| | | 6 | 0.144(0.405) | 0.086(0.125) | 0.066(0.087) | 0.054(0.091) | 0.065(0.093) |
| | | 9 | 0.296(0.469) | 0.233(0.273) | 0.143(0.230) | 0.117(0.158) | 0.112(0.173) |
| 50 | 0 | 0 | 0.009(0.018) | 0.006(0.008) | 0.003(0.004) | 0.004(0.005) | 0.004(0.006) |
| | | 3 | 0.037(0.065) | 0.027(0.036) | 0.018(0.027) | 0.016(0.019) | 0.014(0.024) |
| | | 6 | 0.083(0.139) | 0.063(0.086) | 0.046(0.065) | 0.034(0.043) | 0.028(0.032) |
| | | 9 | 0.093(0.135) | 0.109(0.189) | 0.078(0.110) | 0.055(0.068) | 0.073(0.120) |
| | 1 | 0 | 0.010(0.025) | 0.005(0.008) | 0.005(0.006) | 0.005(0.008) | 0.005(0.008) |
| | | 3 | 0.036(0.047) | 0.031(0.040) | 0.021(0.031) | 0.015(0.021) | 0.012(0.018) |
| | | 6 | 0.091(0.144) | 0.062(0.078) | 0.043(0.062) | 0.035(0.040) | 0.033(0.048) |
| | | 9 | 0.188(0.319) | 0.121(0.164) | 0.118(0.165) | 0.064(0.079) | 0.082(0.125) |

where $\epsilon_{i,t,1}, \epsilon_{i,t,2}, \epsilon_{i,t,3}$ and $\xi_{i,t}$ are form $N(0,1)$. We choose $n = 50$, $T \in \{10, 30, 50\}$, $\delta_s \in \{0, 1\}$, and $\delta \in \{0, 3, 6, 9\}$. The true average treatment effect is also calculated by the difference in the empirical average of rewards between the treatment and the control groups using a sample of 1000 trajectories.

We present means and standard deviations of the MSEs in Table 2. Figure S2 presents additional boxplots of the MSEs for $\delta_s = 1$ and $T \in \{10, 30\}$. These results show similar patterns to those in Example 5.1. The proposed methods show superior performance over the other three methods in almost all scenarios.

Furthermore, we empirically explore the impact of the burn-in period. We set $n = 50$ and $T = 10$. Table 3 reports the Monte Carlo averages of the MSEs with the estimated ATE, along with the corresponding standard errors in parentheses, with different burn-in periods given by $2m_0$. As the

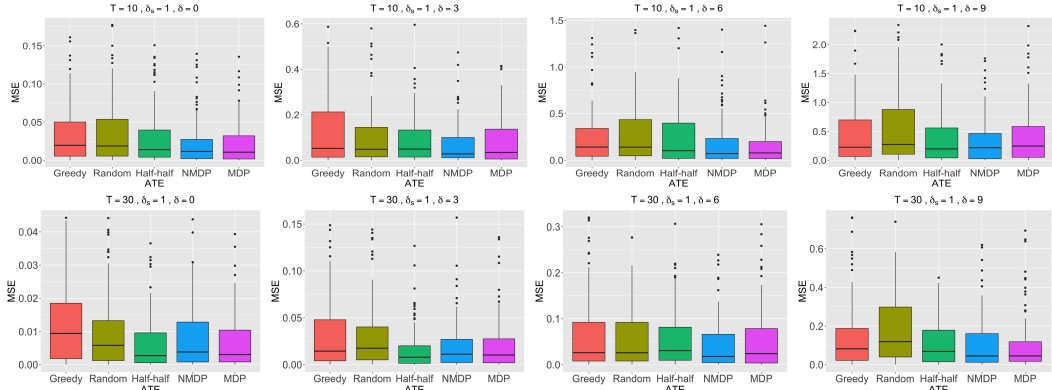

Figure S2: Boxplots of the MSEs of the allocation methods corresponding to $\delta_s = 1$ and $T = \in \{10, 30\}$ in Example 5.2: the four panels correspond to $\delta = 0, 3, 6, 9$, respectively.

Table 3: Simulation results with different burn-in periods for Example 5.2 of Monte Carlo averages with standard errors in parentheses of MSEs for the average treatment effect based on 200 replicates.

| $\delta_s$ | $\delta$ | $2m_0 = 4$ | $2m_0 = 8$ | $2m_0 = 12$ | $2m_0 = 16$ | $2m_0 = 20$ | $2m_0 = 24$ |
|---|---|---|---|---|---|---|---|
| 0 | 0 | 1.754(8.071) | 0.123(0.383) | 0.151(0.687) | 0.061(0.134) | 0.099(0.549) | 0.023(0.031) |
| | 3 | 1.470(6.832) | 1.256(9.471) | 0.183(0.342) | 0.300(1.560) | 0.097(0.141) | 0.088(0.170) |
| | 6 | 2.530(7.354) | 0.586(1.072) | 0.413(0.918) | 0.832(3.976) | 0.219(0.358) | 0.188(0.226) |
| | 9 | 2.057(5.089) | 1.773(4.024) | 0.913(3.754) | 0.685(1.942) | 0.402(0.593) | 0.306(0.423) |
| 1 | 0 | 0.841(4.031) | 0.129(0.384) | 0.452(3.718) | 0.098(0.369) | 0.048(0.069) | 0.026(0.037) |
| | 3 | 1.476(8.593) | 0.375(1.037) | 0.207(0.434) | 0.154(0.570) | 0.093(0.133) | 0.080(0.118) |
| | 6 | 3.742(14.735) | 0.644(1.558) | 0.399(0.687) | 0.326(0.942) | 0.175(0.243) | 0.209(0.349) |
| | 9 | 3.728(10.581) | 1.812(5.619) | 0.610(0.936) | 0.582(0.936) | 0.647(3.689) | 0.342(0.413) |

table indicates, smaller burn-in periods (e.g., $2m_0 = 4$) result in larger MSEs for the estimated ATE. This suggests that inaccurate initial estimators of $V$ and $\sigma^2$ can lead to unstable ATE estimators. To the contrary, when the burn-in period is moderately large (e.g., $2m_0 = 16$ or 20), the estimated ATE becomes more stable, with much smaller MSEs and standard errors. This experiment demonstrates the necessity and impact of the initial burn-in period.

**Example 5.3 (Continued).** Similar to Xu et al. (2018), we examine the interactions between drivers and orders in a $9 \times 9$ spatial grid with a duration of 20 time steps. Drivers are constrained to move vertically or horizontally by only one grid at each time step, while orders can only be dispatched to drivers within a Manhattan distance of 2. An order will be canceled if not being assigned to any driver for a long time. The cancellation time follows a truncated Gaussian distribution with a mean of 2.5 and a standard deviation of 2, ranging from 0 to 5 on the temporal axis. To generate realistic traffic patterns that mimic a morning peak and a night peak, we model residential and working areas separately, and orders' starting locations are sampled using a two-component Gaussian mixture distribution. The locations are then truncated to integers within the spatiotemporal grid. Orders' destinations and drivers' initial locations are randomly sampled from a discrete uniform distribution on the grid. The parameters of the mixture of Gaussians are as follows.

$$\pi^{(1)} = 1/3, \ \pi^{(1)} = 2/3, \ \mu^{(1)} = [3, 3, 2], \ \mu^{(2)} = [6, 6, 15], \ \sigma^{(1)} = [2, 2, 3], \ \sigma^{(2)} = [2, 2, 3].$$

The three dimensions correspond to the spatial horizontal and vertical coordinates, and the temporal coordinate respectively. The true average treatment effect is also calculated by the difference in the empirical average of revenue between the two policies using a sample of 1000 episodes.

Table 4 gives the simulation results of the Monte Carlo averages with standard errors of MSEs. Figure S3 presents addition boxplots of MSEs for $n = 100$ with different numbers of drivers. The corresponding results show that the proposed treatment allocation methods have better performance than the other methods in almost all cases.

Table 4: Simulation results for Example 5.3 of Monte Carlo averages with standard errors in parentheses of MSEs for the average treatment effect based on 200 replicates.

| Drivers | $n$ | Greedy | Random | Half-half | NMDP | TMDP |
|---|---|---|---|---|---|---|
| U(25,30) | 30 | 0.940(1.451) | 0.884(1.119) | 0.158(0.219) | 0.261(0.304) | 0.216(0.324) |
| | 50 | 0.741(0.890) | 0.548(0.842) | 0.178(0.206) | 0.159(0.206) | 0.188(0.294) |
| | 100 | 0.313(0.463) | 0.193(0.281) | 0.059(0.071) | 0.078(0.103) | 0.087(0.126) |
| 25 | 30 | 0.954(1.494) | 0.856(0.912) | 0.246(0.356) | 0.220(0.370) | 0.221(0.333) |
| | 50 | 0.723(0.902) | 0.611(0.766) | 0.136(0.198) | 0.123(0.212) | 0.165(0.223) |
| | 100 | 0.424(0.522) | 0.328(0.443) | 0.083(0.092) | 0.069(0.101) | 0.062(0.083) |
| 50 | 30 | 1.261(1.788) | 1.357(1.825) | 0.486(0.674) | 0.387(0.509) | 0.304(0.451) |
| | 50 | 0.756(1.009) | 0.698(0.932) | 0.266(0.329) | 0.230(0.296) | 0.188(0.260) |
| | 100 | 0.438(0.661) | 0.520(0.696) | 0.123(0.169) | 0.111(0.147) | 0.111(0.155) |

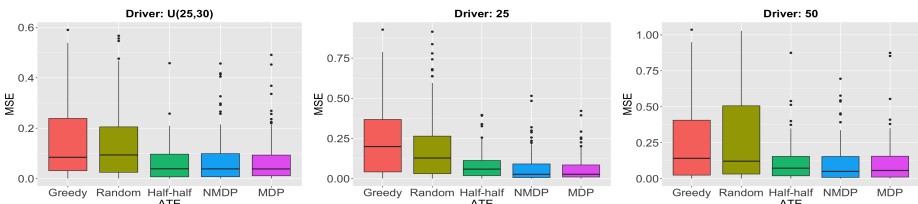

Figure S3: Boxplots of the MSEs of the allocation methods in Example 5.3 with $n = 100$: the three panels correspond to drivers generated from $U(25, 30)$ or fixed as 25 and 50, respectively.

**Example 5.4 (Continued).** In this example, we evaluate the proposed treatment allocation method on a dispatch simulator based on order-driver historical dataset from a world-leading ride-sharing platform. The dataset contains historical data in city A and consists of four parts: the driver trajectory dataset, the trip request dataset, the driver random walk dataset, and driver's online-offline dataset. Each part of data consists both temporal spatial information of orders or drivers and numerical features of them. For example, the trip request data consists the ride start and end time, the pickup and drop-off locations, whether the request is answered by a driver and the Gross Merchandise Volume(GMV) of the request.

We apply the proposed method to the simulator used in Tang et al. (2019) to generate the data based on the historical data, which is a simplified procedure to model the real-world ride-sharing market. To be specific, we first divide the city into $N$ hexagons and split one day into $T$ time slices. At the beginning of each episode, the simulator is initialized by setting the distribution of drivers over the city identical to historical distribution. For each time slice, driver updating and order updating are two main tasks of the simulator. The driver updating task consists of five parts: Some idle drivers who are assigned orders will take the order with some probability, which is calculated by a pretrained light-gbm model given information of driver and the order. Idle drivers who missed orders will head towards some locations determined by the driver random walk dataset. Reposed drivers take the repositioning command from platform based on some pretrained repositioning algorithm. Busy drivers who have already taken orders would go to the appointed locations, pick up the passenger and head towards the destination. New drivers come into this city or old drivers get offline according to the online-offline dataset. Above parts will change the position of certain drivers, resulting in the variation of driver distribution. As for the order updating, there exist two major parts: new requests are generated according to the request dataset. The unassigned orders and new requests are transformed into the formal order and dispatched to idle drivers.

We set $T = 20$ for the order dispatch that would be conducted at each hour and the platform work time ranges from 4:00 to 24:00, $N = 85$ for the geographic reason, and $n$ to be 7 days or 10 days. The true average treatment effect is calculated by the difference in the empirical average of GMV between the two policies over 100 days.

## S3 Proofs of the main theorems

*Proof of Theorem 1.* Recall that in NMDPs, for a general behavior policy $\pi^b$, the efficiency bound is

$$\text{EB}_1(\pi^b) = \sum_{t=1}^{T} \underbrace{\frac{1}{T^2} \sum_{a \in \{0,1\}} \mathbb{E}^{\pi^b} \Big[ \sigma_t(H_t, a) \prod_{k \leq t} \frac{\mathbb{I}(A_k = a)}{\pi_k^b(a|H_k)} \Big]^2}_{\text{EB}_1^{(t)}(\{\pi_j^b\}_{j \leq t})} + \frac{1}{T^2} \text{Var}[V_1^1(O_1) - V_1^0(O_1)].$$

As commented earlier, the second term is independent of $\pi^b$. It suffices to search the optimal behavior policy $\pi^{b*}$ that minimizes the first term, or equivalently, $\sum_{t=1}^{T} \text{EB}_1^{(t)}(\{\pi_j^b\}_{j \leq t})$.

We identify $\pi^{b*}$ in a recursive manner. We start with $\pi_T^{b*}$. Notice that $\pi_T^{b*}$ is independent of $\sum_{t<T} \text{EB}_1^{(t)}(\{\pi_j^b\}_{j \leq t})$, it suffices to identify $\pi_T^{b*}$ that minimizes $\text{EB}_1^{(T)}(\pi^b)$. Using the law of iterated expectations, $\text{EB}_1^{(T)}(\pi^b)$ can be rewritten as

$$\frac{1}{T^2} \sum_{a \in \{0,1\}} \mathbb{E}\Big[ \prod_{k \leq T-1} \frac{\mathbb{I}(A_k = a)}{\pi_k^b(a|H_t)^2} \mathbb{E}\Big\{ \frac{\mathbb{I}(A_T = a)}{\pi_T^b(a|H_T)^2} \sigma_T^2(H_T, a)|H_T \Big\} \Big]$$

$$= \frac{1}{T^2} \sum_{a \in \{0,1\}} \mathbb{E}\Big[ \frac{\mathbb{I}(A_1 = \ldots A_{T-1} = a)}{\prod_{k \leq T-1} \pi_k^b(a|H_t)^2} \frac{\sigma_T^2(H_T, a)}{\pi_T^b(a|H_T)} \Big].$$

A key observation is that, both the variable inside the square brackets on the second line and $\pi_T^b$ itself are functions of $H_T$. Therefore, it suffices to identify $\pi_T^{b*}$ that minimizes the following conditional expectation

$$\frac{1}{T^2} \sum_{a \in \{0,1\}} \mathbb{E}\Big[ \frac{\mathbb{I}(A_1 = \ldots A_{T-1} = a)}{\prod_{k \leq T-1} \pi_k^b(a|H_t)^2} \frac{\sigma_T^2(H_T, a)}{\pi_T^b(a|H_T)} \Big| H_T \Big].$$

When $H_T$ is given, $A_1, \cdots, A_{T-1}$ are fixed as well. As such, we have either $A_1 = \cdots = A_{T-1} = 0$ or $A_1 = \cdots = A_{T-1} = 1$, but not necessarily both. It suffices to set $\pi_T^{b*}(a|H_T) = 1$ whenever $\mathbb{I}(A_1 = \cdots = A_{T-1} = a)$ for some value of $a$. Notice that this is automatically guaranteed when we set $\pi_T^{b*}(A_1|H_T) = 1$.

So far we have specified the form of $\pi_T^{b*}$. We next identify $\pi_{T-1}^{b*}$. Similarly, $\pi_{T-1}^{b*}$ is independent of $\sum_{t<T-1} \text{EB}_1^{(t)}(\{\pi_j^b\}_{j \leq t})$. It suffices to consider $\pi_{T-1}^{b*}$ that minimizes $\text{EB}_1^{(T-1)}(\{\pi_j^b\}_{j \leq T-1}) + \text{EB}_1^{(T)}(\{\pi_j^b\}_{j \leq T-1} \cup \pi_T^{b*})$. Using the law of iterated expectations again, we can show that $\text{EB}_1^{(T-1)}(\{\pi_j^b\}_{j \leq T-1}) + \text{EB}_1^{(T)}(\{\pi_j^b\}_{j \leq T-1} \cup \pi_T^{b*})$ equals

$$\sum_{a \in \{0,1\}} \mathbb{E}\Big[ \frac{\mathbb{I}(A_1 = \ldots A_{T-1} = a)}{\prod_{k \leq T-1} \pi_k^b(a|H_t)^2} \mathbb{E}[\sigma_{T-1}^2(H_{T-1}, a) + \{\sigma_T^2(H_T, a)|H_{T-1}, A_{T-1} = a\}] \Big]$$

$$= \sum_{a \in \{0,1\}} \mathbb{E}\Big[ \frac{\mathbb{I}(A_1 = \ldots A_{T-2} = a)}{\prod_{k \leq T-2} \pi_k^b(a|H_t)^2} \frac{\mathbb{E}[\sigma_{T-1}^2(H_{T-1}, a) + \{\sigma_T^2(H_T, a)|H_{T-1}, A_{T-1} = a\}]}{\pi_{T-1}^b(a|H_{T-1})} \Big].$$

Notice that the second line is again a function of $H_{T-1}$. Using similar arguments, we can show that $\pi_{T-1}^{b*}$ satisfies $\pi_{T-1}^{b*}(A_1|H_{T-1}) = 1$. Very similarly, we can prove $\pi_{T-2}^{b*}(A_1|H_{T-2}) = \cdots = \pi_2^{b*}(A_1|H_2) = 1$ in a sequential order.

Finally, we need to identify $\pi_1^{b*}$ that minimizes $\text{EB}_1^{(1)}(\pi_1^b) + \sum_{1 < t \leq T} \text{EB}_1^{(t)}(\pi_1^b \cup \{\pi_j^{b*}\}_{1 < j \leq t})$, which equals

$$\frac{1}{T^2} \sum_{a=0}^{1} \mathbb{E}\Big[ \frac{\sum_{t=1}^{T} \mathbb{E}\{\sigma_t^2(H_t, a)|A_1 = \cdots = A_{T-1} = a, O_1\}}{\pi_1^b(a|O_1)} \Big].$$

Notice that the numerator inside the square brackets is equal to $\sigma_*^2(O_1, a)$. It suffices to solve the following constrained optimization,

$$\min_{\pi_1^b} \mathbb{E}\Big[ \frac{\sigma_*^2(O_1, 1)}{\pi_1^b(1|O_1)} + \frac{\sigma_*^2(O_1, 0)}{\pi_1^b(0|O_1)} \Big], \quad \text{s.t.} \quad \pi_1^b(1|O_1) + \pi_1^b(0|O_1) = 1.$$

or equivalently,

$$\min_{\pi_1^b} \mathbb{E}\left[\frac{\sigma_*^2(O_1, 1)}{\pi_1^b(1|O_1)} + \frac{\sigma_*^2(O_1, 0)}{\pi_1^b(0|O_1)}\bigg| O_1\right], \quad \text{s.t.} \quad \pi_1^b(1|O_1) + \pi_1^b(0|O_1) = 1,$$

This leads to

$$\pi_1^b(a|O_1) = \frac{\sigma_*(O_1, a)}{\sigma_*(O_1, 0) + \sigma_*(O_1, 1)}.$$

The proof is hence completed. □

*Proof of Theorem 2.* For any $\pi_1^b$ and $V_1^a$, define

$$\psi(\pi_1^b, V_1^a, O_1) = V_1^a(O_1) + \frac{\mathbb{I}(A_1^{(i)} = a)}{\pi_1^b(a|O_1)}[\sum_t R_t - V_1^a(O_1)].$$

Notice that $A_1^{(i)}$ is generated according to $\widehat{\pi}_{1,i-1}^{b*}$. As such, $\psi(\widehat{\pi}_{1,i-1}^{b*}, \widehat{V}_{1,i-1}^a, O_1^{(i)})$ is unbiased to the oracle value and $\widehat{\text{ATE}}_1$ is unbiased to $\text{ATE}_1$. This allows us to represent

$$\widehat{\text{ATE}}_1 - \text{ATE}_1 = \frac{1}{(n - 2m_0)T} \sum_{a\in\{0,1\}} (-1)^{a+1} \sum_{i=2m_0+1}^n \left[\psi(\widehat{\pi}_{1,i-1}^{b*}, \widehat{V}_{1,i-1}^a, O_1^{(i)}) - \mathbb{E}\psi(\widehat{\pi}_1^{b*}, V_1^a, O_1^{(i)})\right].$$

Let $\mathcal{D}_i$ denote the data up to the $i$-th day. A key observation is that, $\widehat{\text{ATE}}_1 - \text{ATE}_1$ corresponds to a sum of martingale differences with respect to the filtration $\{\mathcal{D}_i\}_i$. These martingale differences are uncorrelated. As such,

$$\mathbb{E}(\widehat{\text{ATE}}_1 - \text{ATE}_1)^2$$

$$= \frac{1}{(n - 2m_0)^2 T^2}\mathbb{E}\left\{\sum_{i=2m_0+1}^n \sum_{a\in\{0,1\}} (-1)^{a+1}\left[\psi(\widehat{\pi}_{1,i-1}^{b*}, \widehat{V}_{1,i-1}^a, O_1^{(i)}) - \mathbb{E}\psi(\widehat{\pi}_1^{b*}, V_1^a, O_1^{(i)})\right]\right\}^2$$

$$= \frac{1}{(n - 2m_0)^2 T^2}\sum_{i=2m_0+1}^n \mathbb{E}\left\{\sum_{a\in\{0,1\}} (-1)^{a+1}\left[\psi(\widehat{\pi}_{1,i-1}^{b*}, \widehat{V}_{1,i-1}^a, O_1^{(i)}) - \mathbb{E}\psi(\widehat{\pi}_1^{b*}, V_1^a, O_1^{(i)})\right]\right\}^2$$

With some calculations, we can decompose the variable inside the curly brackets into the sum of $\psi_{1,i} + \psi_{2,i} + \psi_{3,i}$ where $\psi_{1,i} = V_1^1(O_1^{(i)}) - V_1^0(O_1^{(i)}) - \text{ATE}_1$,

$$\psi_{2,i} = \sum_{a\in\{0,1\}} (-1)^{a+1}\frac{\mathbb{I}(A_1^{(i)} = a)}{\widehat{\pi}_1^{b*}(a|O_1)}[\sum_t R_t - V_1^a(O_1)],$$

$$\psi_{3,i} = \sum_{a\in\{0,1\}} (-1)^{a+1}\left[1 - \frac{\mathbb{I}(A_1^{(i)} = a)}{\widehat{\pi}_1^{b*}(a|O_1^{(i)})}\right]\left[\widehat{V}_{1,i-1}^a(O_1^{(i)}) - V^a(O_1^{(i)})\right].$$

It is straightforward to show that $\psi_{1,i}$, $\psi_{2,i}$ and $\psi_{3,i}$ are uncorrelated. Hence,

$$\mathbb{E}(\widehat{\text{ATE}}_1 - \text{ATE}_1)^2 = \frac{1}{(n - 2m_0)^2 T^2}\sum_{i=2m_0+1}^n [\mathbb{E}\psi_{1,i}^2 + \mathbb{E}\psi_{2,i}^2 + \mathbb{E}\psi_{3,i}^2]$$

$$= \frac{\text{EB}_1(\widehat{\pi}^{b*})}{n - 2m_0} + \frac{1}{(n - 2m_0)^2 T^2}\sum_{i=2m_0+1}^n \mathbb{E}\psi_{3,i}^2. \quad \text{(S3)}$$

To establish Theorem 2, we need to first upper bound the difference between $\text{EB}_1(\widehat{\pi}^{b*})$ and the minimal efficiency bound $\text{EB}_1(\pi^{b*})$. Under the conditions that $\max_{a,o} \sigma_*^2(o, a) \leq T^2 R_{\max}^2$, we obtain

$$|\text{EB}_1(\widehat{\pi}^{b*}) - \text{EB}_1(\pi^{b*})| \leq \frac{1}{(n - 2m_0)T^2}\sum_{a\in\{0,1\}} \sum_{i=2m_0+1}^n \mathbb{E}\left|\frac{\sigma_*^2(O_1^{(i)}, a)}{\widehat{\pi}_{1,i-1}^{b*}(a|O_1^{(i)})} - \frac{\sigma_*^2(O_1^{(i)}, a)}{\pi^{b*}(a|O_1^{(i)})}\right|$$

$$\leq \frac{2R_{\max}^2}{n - 2m_0}\sum_{i=2m_0+1}^n \mathbb{E}\left|\frac{1}{\widehat{\pi}_{1,i-1}^{b*}(a|O_1^{(i)})} - \frac{1}{\pi_1^{b*}(a|O_1^{(i)})}\right|. \quad \text{(S4)}$$

Under the conditions on the estimated behavior policy $\widehat{\pi}^{b*}_{1,i-1}$, it follows from Cauchy-Schwarz inequality that

$$\mathbb{E}\Big|\frac{1}{\widehat{\pi}^{b*}_{1,i}(a|O_1)} - \frac{1}{\pi^{b*}_1(a|O_1)}\Big| \leq \Big\{\mathbb{E}\Big[\frac{1}{\widehat{\pi}^{b*}_{1,i}(a|O_1)} - \frac{1}{\pi^{b*}_1(a|O_1)}\Big]^2\Big\}^{1/2} = \sqrt{C}i^{-\alpha_1}.$$

Notice that

$$\frac{1}{n-2m_0}\sum_{i=2m_0+1}^{n}(i-1)^{-\alpha_1} \leq \frac{1}{n-2m_0}\sum_{i=1}^{n-2m_0}i^{-\alpha_1} \leq \frac{1}{n-2m_0}\Big[1+\int_{1}^{n-2m_0-1}x^{-\alpha_1}dx\Big]$$

$$\leq (n-2m_0)^{-1} + \frac{1}{1-\alpha_1}(n-2m_0)^{-\alpha_1} \leq \frac{2-\alpha_1}{1-\alpha_1}(n-2m_0)^{-\alpha_1}. \tag{S5}$$

By (S4), it yields the following upper bound for $(n-2m_0)^{-1}[\text{EB}_1(\widehat{\pi}^{b*}) - \text{EB}_1(\pi^{b*})]$:

$$\frac{2(2-\alpha_1)\sqrt{C}R^2_{\max}}{(1-\alpha_1)(n-2m_0)^{1+\alpha_1}}. \tag{S6}$$

It remains to upper bound the second term in (S3). Under the condition that $\min_{1,i}\widehat{\pi}^{b*}_i \geq \epsilon$, we obtain that

$$\mathbb{E}\psi^2_{3,i} \leq 2\sum_{a=0}^{1}\mathbb{E}\Big\{\Big[\widehat{V}^a_{1,i-1}(O_1^{(i)}) - V^a(O_1^{(i)})\Big]^2\mathbb{E}\Big[\Big(1-\frac{\mathbb{I}(A_1^{(i)}=a)}{\widehat{\pi}^{b*}_1(a|O_1^{(i)})}\Big)^2|\mathcal{D}_{i-1},O_1^{(i)}\Big]\Big\}$$

$$\leq 2\epsilon^{-1}\sum_{a=0}^{1}\mathbb{E}\Big[\widehat{V}^a_{1,i-1}(O_1^{(i)}) - V^a(O_1^{(i)})\Big]^2 \leq 2C\epsilon^{-1}i^{-2\alpha_2}T^2.$$

Similar to (S5), we can show that the second term in (S3) is of the order of magnitude $O(C\epsilon^{-1}(n-2m_0)^{-1-2\alpha_2})$. Plugging this and (S6) into (S3) yields the desired results. $\qquad\square$

*Proof of Proposition 1.* We give a counterexample that the optimal behavior policy $\pi^{b*}$ does not belong to the set $\Pi^b$.

We design a two-stage TMDP $\{O_t, A_1, R_t\}^2_{t=1}$ with binary observations. Assume $O_1 = R_1 = 0$ almost surely. The distribution of $O_2$ depends on $A_1$. In particular, let $\mathbb{P}(O_2 = 1|A_1 = 1) = 0.5$ and $\mathbb{P}(O_2 = 1|A_1 = 0) = p$ for some $0 < p < 1$. In addition, let $R_2 = O_2 + e_2$ for some error term $e_2$ such that $\mathbb{E}(e_2|A_2, O_2) = 0$. It is immediate to see that $\sigma^2_2$ corresponds to the conditional variance of $e_2$ given $(O_2, A_2)$. Meanwhile, we have $V^a_2(o_2) = o_2$ for any $a$ and $o_2$. Since $O_1$ is degenerate, we write $\sigma_1(O_1, A_1)$ as $\sigma_1(A_1)$. It is straightforward to show that $\sigma_1(1) = \text{Var}(O_2|A_1 = 1) = 0.5, \sigma_1(0) = \text{Var}(O_2|A_1 = 0) = \sqrt{p(1-p)}$.

Equation (S12) provides the closed-form expressions for the marginal probability mass functions $p^b_1$ and $p^b_2$ under $\pi^{b*}$, provided that there exists $\pi^{b*}$ such that the induced marginal probability mass functions match these distribution functions. In particular, according to (S12),

$$p^b_1(a) = \frac{\sigma_1(a)}{0.5+\sqrt{p(1-p)}} \quad\text{and}\quad p^b_2(o,a) = \frac{p^a_2(o)\sigma_2(o,a)}{\sum_{a',o'}p^{a'}_2(o')\sigma_2(o',a')}, \tag{S7}$$

where $p^b_1$ is a function of the initial action only, due to the degeneracy of the initial observation. By direct calculations, we obtain that $p^1_2(1) = 0.5, p^0_2(1) = p$.

We next establish the existence of $\pi^{b*}$ such that the induced $p^b_1$ and $p^b_2$ equal (S7). This proves that $\pi^{b*}$ is indeed a minimizer of $\text{EB}_2$. Meanwhile, we show that any minimizer of $\text{EB}_2$ does not belong to $\Pi^b$. This would complete the proof.

The identification of $\pi^{b*}_1$ is straightforward as $O_1$ is degenerate. We can simply set $\pi^{b*}_1$ to $p^b_1$. It remains to specify $\pi^{b*}_2$. By definition,

$$p^b_2(o_2, a_2) = \sum_{a_1\in\{0,1\}}\pi^{b*}_1(a_1)\mathbb{P}(O_2 = o_2|A_1)\pi^{b*}_2(a_2|a_1,o_2)$$

$$= \sum_{a_1\in\{0,1\}}p^b_1(a_1)\mathbb{P}(O_2 = o_2|A_1)\pi^{b*}_2(a_2|a_1,o_2).$$

By summing over $a_2$ on both sides, we obtain the constraints that

$$p_2^b(o_2, 1) + p_2^b(o_2, 0) = \sum_{a_1 \in \{0,1\}} p_1^b(a_1) \mathbb{P}(O_2 = o_2 | A_1),$$

for any $o_2$, or equivalently,

$$
\begin{aligned}
p_2^b(0,1) + p_2^b(0,0) = \quad & 0.5 \frac{1}{1 + 2\sqrt{p(1-p)}} + (1-p)\frac{2\sqrt{p(1-p)}}{1 + 2\sqrt{p(1-p)}}, \\
p_2^b(1,1) + p_2^b(1,0) = \quad & 0.5 \frac{1}{1 + 2\sqrt{p(1-p)}} + p\frac{2\sqrt{p(1-p)}}{1 + 2\sqrt{p(1-p)}}.
\end{aligned}
\tag{S8}
$$

Notice that at this stage, we have not specified $\sigma_2$ yet. For a given $0 < p < 1$, $p_2$ can be any strictly positive probability mass function by adjusting the values of $\sigma_2$. A extreme case is given by $p = 1/2$. In this case, $O_2$ is independent of $A_1$ and we can set $\sigma_2$ to a constant to ensure (S8) holds. Then the optimal $\pi_2^{b*}(a_2|a_1, o_2) = 0.5$ regardless of the values of $a_1, a_2$ and $o_2$. Apparently, $\pi^{b*} \notin \Pi^b$ in this case.

We consider a more complicated case where $p > 0.5$. As commented earlier, $p_2^b$ can be any strictly positive probability mass function by adjusting the values of $\sigma_2$. Toward that end, we set $\sigma_2$ to be such that $p_2^b(1,1) = 0.25$, $p_2^b(0,1) = 0.5 - 0.5p$, and the values of $p_2^b(1,0)$, $p_2^b(0,0)$ can be calculated according to (S8). When $p > 0.5$, the second term on the right-hand-side (RHS) of the second line is larger than 0.5. As such, $p_2^b(1,0)$ is well-defined. Similarly, we can show that $p_2^b(0,0)$ is well-defined.

It remains to specify $\pi_2^{b*}$ to satisfy the following two equations:

$$p_2^b(1,1) = 0.25 = 0.5 \frac{1}{1 + 2\sqrt{p(1-p)}} \pi_2^{b*}(1|1,1) + p\frac{2\sqrt{p(1-p)}}{1 + 2\sqrt{p(1-p)}} \pi_2^{b*}(1|0,1), \tag{S9}$$

$$p_2^b(0,1) = 0.5 - 0.5p = 0.5 \frac{1}{1 + 2\sqrt{p(1-p)}} \pi_2^{b*}(1|1,0) + (1-p)\frac{2\sqrt{p(1-p)}}{1 + 2\sqrt{p(1-p)}} \pi_2^{b*}(1|0,0). \tag{S10}$$

When these two equations are satisfied, the constraints on $p_2^b(1,0)$ and $p_2^b(0,0)$ are automatically satisfied due to (S8).

Consider the first equation (S9). When $p > 0.5$, the weight $0.5/(1 + 2\sqrt{p(1-p)})$ is smaller than 0.25 whereas the weight $2p\sqrt{p(1-p)}/(1 + 2\sqrt{p(1-p)})$ is larger than 0.25. So there must exist $\pi_2^{b*}(1|1,1)$ and $\pi_2^{b*}(1|0,1)$ that satisfy this equation. Meanwhile, when $\pi_2^{b*}$ sets $A_2 = A_1$, i.e., $\pi_2^{b*}(1|1,1) = 1$ and $\pi_2^{b*}(1|0,1) = 0$, the RHS is strictly smaller than 0.25. This shows that any $\pi^b \in \Pi^b$ will not satisfy (S9).

As for the second equation, similarly, the weight $2(1-p)\sqrt{p(1-p)}/(1 + 2\sqrt{p(1-p)})$ is larger than $0.5 - 0.5p$. Hence, there must exist $\pi_2^{b*}(1|1,0)$ and $\pi_2^{b*}(1|0,0)$ that satisfy this equation. The proof is hence completed.

$\square$

*Proof of Theorem 3.* We first prove the *in-class* asymptotic optimality under the $\beta$-mixing condition. In other words, we aim to show that $\pi^{b*}$ asymptotically minimizes

$$\min_{\pi^b \in \Pi^b} \frac{1}{T} \sum_{t=1}^{T} \left[ \mathbb{E} \left\{ \frac{\mathbb{I}(A_t = 1) p_t^1(O_t)^2}{p_t^b(O_t, 1)^2} \sigma_t^2(O_t, 1) \right\} + \mathbb{E} \left\{ \frac{\mathbb{I}(A_t = 0) p_t^0(O_t)^2}{p_t^b(O_t, 0)^2} \sigma_t^2(O_t, 0) \right\} \right].$$

By definition, $p_t^b(o_t, a) = \int \pi_1^b(a|o_1) p_t^a(o_t|o_1) p(o_1) do_1$. Under the $\beta$- mixing condition, $p_t^b(o_t, a)$ converges to $\pi_1^b(a) p_t^a(o_t)$ uniformly in $a$ and $o_t$, where $\pi_1^b(a) = \mathbb{E}\pi_1^b(a|O_1)$. The above minimization problem is asymptotically equivalent to

$$\min_{\pi_1^b} \frac{1}{T} \sum_{t=1}^{T} \left[ \mathbb{E} \left\{ \frac{\mathbb{I}(A_1 = 1)}{\pi_1^b(1)^2} \mathbb{E}^1 \sigma_t^2(O_t, 1) \right\} + \mathbb{E} \left\{ \frac{\mathbb{I}(A_1 = 0)}{\pi_1^b(0)^2} \mathbb{E}^0 \sigma_t^2(O_t, 0) \right\} \right],$$

Similar to the proof of Theorem 1, under the constraint that $\pi_1^b(1) + \pi_1^b(0) = 1$, the minimum can be achieved by setting

$$\pi_1^{b*}(a) = \frac{\sigma_{a*}}{\sigma_{1*} + \sigma_{0*}}, \sigma_{a*}^2 = \frac{1}{T^2} \sum_{t=1}^{T} \mathbb{E}^a \left[ \sigma_t^2(O_t, a) \right].$$

This proves the first part of Theorem 3.

We next establish the second part. We aim to identify $\pi^b$ that minimizes

$$\frac{1}{T} \min_{\pi^b} \sum_{t=1}^{T} \left[ \mathbb{E}\left\{ \frac{\mathbb{I}(A_t = 1)p_t^1(O_t)^2}{p_t^b(O_t, A_t)^2} \sigma_t^2(O_t, 1) \right\} + \mathbb{E}\left\{ \frac{\mathbb{I}(A_t = 0)p_t^0(O_t)^2}{p_t^b(O_t, A_t)^2} \sigma_t^2(O_t, 0) \right\} \right]. \quad \text{(S11)}$$

Direct calculations lead to

$$\mathbb{E}\left\{ \frac{\mathbb{I}(A_t = a)p_t^a(O_t)^2}{p_t^b(O_t, A_t)^2} \sigma_t^2(O_t, a) \right\} = \sum_o \frac{p_t^a(o)^2}{p_t^b(o, a)^2} \sigma_t^2(o, 1)p_t^b(o, a) = \sum_o \frac{p_t^a(o)^2 \sigma_t^2(o, a)}{p_t^b(o, a)}$$

Directly searching the optimal $\pi^b$ is challenging due to the intricate dependence of $p_t^b$ on $\pi^b$. Instead of searching the optimal $\pi^b$ that minimize (S11), we enlarge the search space to consider a sequence of unrestricted density functions $\{p_t^b\}_t$ (e.g., $p_t^b$ is no longer required to be a $\pi^b$-induced probability density/mass function) that minimize

$$\frac{1}{T} \sum_{t=1}^{T} \sum_o \left[ \frac{p_t^1(o)^2 \sigma_t^2(o, 1)}{p_t^b(o, 1)} + \frac{p_t^0(o)^2 \sigma_t^2(o, 0)}{p_t^b(o, 0)} \right], \quad \text{s.t.} \sum_o [p_t^b(o, 1) + p_t^b(o, 0)] = 1.$$

Similar to the proof of Theorem 1, the minimal value can be achieved by setting

$$p_t^b(o, a) = \frac{p_t^a(o)\sigma_t(o, a)}{\sum_{o'} [p_t^1(o')\sigma_t(o', 1) + p_t^0(o')\sigma_t(o', 0)]}. \quad \text{(S12)}$$

Under the constancy assumption, we have $\sigma_t(o, a) = c^a$ for some any $t$ and $o$. It follows that

$$p_t^b(o, a) = \frac{p_t^a(o)c^a}{\sum_{o'} [p_t^1(o')c^1 + p_t^0(o')c^0]} = \frac{p_t^a(o)c^a}{c^1 + c^0}. \quad \text{(S13)}$$

Notice that $\pi_1^{b*}(a) = c^a/(c^1 + c^0)$. Using similar arguments in the proof of first part, under the $\beta$-mixing condition, we can show that the induced $p_t^b$ is asymptotically equivalent to (S13). This implies that the proposed $\pi^{b*}$ is asymptotically optimal among all behavior policies. The proof is hence completed. $\square$

*Proof of Theorem 4.* The proof of Theorem 4 is very similar to that of Theorem 2. We begin by defining the estimating function

$$\phi(\pi_1^b, V_1^a, O_1^{(i)}) = V_1^a(O_1^{(i)}) + \frac{\mathbb{I}(A_1^{(i)} = a)}{\pi_1^b(a)} \sum_{t=1}^{T} [R_t^{(i)} - V_1^a(O_1^{(i)})],$$

which leads to

$$\widehat{\text{ATE}}_2 - \text{ATE}_2 = \sum_{a \in \{0,1\}} \frac{(-1)^{a+1}}{T(n - 2m_0)} \sum_{i=2m_0+1}^{n} \left[ \phi(\widehat{\pi}_{1,i-1}^{b*}, \widehat{V}_{1,i-1}^a, \widehat{O}_1^{(i)}) - \mathbb{E}\phi(\widehat{\pi}_1^{b*}, V_1^a, O_1^{(i)})) \right].$$

The variable inside the curly brackets can be similarly decomposed into the sum of three uncorrelated variables $\phi_{1,i} + \phi_{2,i} + \phi_{3,i}$ where $\phi_{1,i} = V_1^1(O_1^{(i)}) - V_1^0(O_1^{(i)}) - \text{ATE}_2$,

$$\phi_{2,i} = \sum_{a \in \{0,1\}} (-1)^{a+1} \frac{\mathbb{I}(A_1^{(i)} = a)}{\widehat{\pi}_1^{b*}(a)} [\sum_t R_t - V_1^a(O_1)],$$

$$\phi_{3,i} = \sum_{a \in \{0,1\}} (-1)^{a+1} \left[ 1 - \frac{\mathbb{I}(A_1^{(i)} = a)}{\widehat{\pi}_1^{b*}(a)} \right] \left[ \widehat{V}_{1,i-1}^a(O_1^{(i)}) - V^a(O_1^{(i)}) \right].$$

It follows from the martingale structure that

$$\mathbb{E}(\widehat{\mathrm{ATE}}_2 - \mathrm{ATE}_2)^2 = \frac{1}{(n-2m_0)^2 T^2} \sum_{i=2m_0+1}^{n} [\mathbb{E}\phi_{1,i}^2 + \mathbb{E}\phi_{2,i}^2 + \mathbb{E}\phi_{3,i}^2]$$

$$= \frac{\mathrm{EB}_2(\widehat{\pi}^{b*})}{n-2m_0} + \frac{1}{(n-2m_0)^2 T^2} \sum_{i=2m_0+1}^{n} \mathbb{E}\phi_{3,i}^2. \tag{S14}$$

Similar to the derivation of (S4), the difference of $\mathrm{EB}_2(\widehat{\pi}^{b*})$ and the optimal $\mathrm{EB}_2(\pi^{b*})$ can be upper bounded by

$$|\mathrm{EB}_2(\widehat{\pi}^{b*}) - \mathrm{EB}_2(\pi^{b*})| \leq \frac{1}{(n-2m_0)T^2} \sum_{a\in\{0,1\}} \sum_{i=2m_0+1}^{n} \mathbb{E}\left|\frac{\sigma_*^2(O_1^{(i)},a)}{\widehat{\pi}_{1,i-1}^{b*}(a)} - \frac{\sigma_*^2(O_1^{(i)},a)}{\pi_1^{b*}(a)}\right|$$

$$= O(R_{\max}^2 \sqrt{C}(n-2m_0)^{-\alpha_1}). \tag{S15}$$

Meanwhile, we can similarly show that the second term of (S14) is of the order $O(C\epsilon^{-1}(n-2m_0)^{-1-2\alpha_2})$. The proof is hence completed. $\square$