# OpenReview forum: "Optimal Treatment Allocation for Efficient Policy Evaluation in Sequential Decision Making"
_NeurIPS.cc/2023/Conference — NeurIPS 2023 poster_

### Official Review · Reviewer_bgAF · 2023-06-27

**Soundness:** 3 good
**Presentation:** 3 good
**Contribution:** 3 good
**Rating:** 6
**Confidence:** 1

**Summary:**

The paper designs methods for taking actions in multiple sequential settings (Markovian, time-varying Markovian, non-Markovian) to generate high-quality data for the estimation of average treatment effect.

**Strengths:**

1. Focuses on improving ATE estimation by improving the data collection process, as opposed to improving estimators etc, which is a nice take on this problem.

**Weaknesses:**

None identified.

**Questions:**

1. Theorem 1 and the implementation seems like an "explore then commit" type of a strategy seen in bandits. Is there any connection/parallels?
2. No need for the number of days in Sec 2.
3. It is not clear why running the two fixed policies sequentially one after the other cannot estimate ATE well.
4. Maybe this is easy to see, but Theorem 1's pi* is not taking one of the actions at all (e.g., if A1 happens to be 1, then A2=A1... etc). If so, why would ATE be estimated well?
5. If you are already able to estimate V_1^a in step 3 of Algo 1, can't you estimate ATE well at this point itself?
6. Clarification question: From Theorem 1, it seems like we are picking one of the actions to run for T periods (or "after burn-in" in Alg 1) that has a higher estimated variance in the rewards it obtains. This seems like a natural thing to do to estimate ATE better. Please correct me if I am wrong here.

**Limitations:**

1. Hard to appreciate the optimal treatment strategy introduced in Sec 3, likely because the treatment is a bit terse and skips some useful discussion.
2. The data collection strategy is agnostic to the estimator used later. Is there any further improvement to be had? It would be good to see comments on this, as the estimators are already known in the literature.

---

> ### Author Rebuttal · Authors · 2023-08-09
>
> * **Connections with bandits.** This is an excellent comment! In the following, we discuss the similarities and differences between  the "explore than commit" strategy  in the bandits literature and our proposal.
>
>   - **Similarities**: In the "explore then commit" strategy, a system initially navigates through a spectrum of alternatives in an exploration phase. Subsequently, based on the data gathered, it commits to the seemingly most promising option during the commitment phase. For reference, see Garivier et al. (2016, https://proceedings.neurips.cc/paper/2016/hash/ef575e8837d065a1683c022d2077d342-Abstract.html). Our proposed experimental design resonates with this approach. The burn-in period can be likened to the "exploration" phase where treatments are alternately assigned to accumulate preliminary data. Post this period, the algorithm decisively adopts the optimal design, aiming to reduce the MSE of the ATE estimator.
>
>   - **Differences**: While both strategies share conceptual similarities, their specific objectives and implementation nuances vary considerably. The "explore then commit" strategy is tailored to maximize the cumulative expected reward over a period, whereas our method is centered on achieving the most precise ATE estimation.
>
> Section 2 - Number of Days: Your insightful recommendation is appreciated! We intend to omit this detail from Section 2 and introduce it within the Implementation subsection of Section 3.
>
> * **Estimate ATE based on two fixed policies running sequentially.** Thank you for highlighting that! Indeed, executing the two fixed policies in sequence can produce consistent ATE estimators. Nonetheless, this resulting estimator might be vulnerable to carryover bias as described by Hu and Wager (2022, https://arxiv.org/abs/2209.00197). Furthermore, it's less efficient compared to the estimator derived from our proposed design, which is crafted to minimize the MSE of the ATE estimator.
>
> * **Reason for well ATE estimates under Theorem 1.** Apologize for the confusion! Although the proposed optimal behavior policy $\pi_1^{b*}$ selects the same action within each day, the experiment extends over a total of $n$ days and the first action is randomly chosen from $\{0,1\}$ according to $\sigma_*$ (see Equation 2). Hence, the two policies will be implemented over approximately $\Omega(n)$ days on average, ensuring consistent estimation of the ATE.
>
> * **Estimate ATE in step 3 of Algo 1.** Many thanks for the valuable comment! Although we can estimate $V\_1^a$ and ATE in step 3 of Algorithm 1, it only uses data collected from the first $2 m\_0 $ days. Since the experiments last for $n > 2 m\_0$ days, our proposal uses all  the data to estimate ATE, which is more efficient.
>
> * **Clarification on Theorem 1.** You've hit the nail on the head! In Theorem 1, we allocate the action with a greater estimated variance a higher likelihood. This concept is inspired by similar strategies found in the design of clinical trials, as exemplified by Yin and Zhou (2017, https://www.jstor.org/stable/26384103). Yet, as mentioned in the introduction, the majority of current literature focuses on single-stage studies. Our proposed work broadens this concept to a dynamic context, where treatments are sequentially assigned over time.
>
> * **Optimal treatment strategy in Sec 3.** We sincerely apologize for any confusion and the lack of clarity. To address your concerns, we provide a non-technical summary of our proposal.
>
>   - **Objective:** Our focus is to explore the optimal dynamic treatment allocation strategy for collecting experimental data spanning $n$ days, each containing $T$ time intervals. The goal is to maximize the information (i.e., to minimize the MSE) of the ATE estimator derived from this collected dataset.
>   - **Optimality Criterion:**  To maximize the estimation accuracy, we focus on the class of semiparametrically efficient estimators, whose MSEs attain the semiparametric efficiency bounds.  Hence, it suffices to identify the optimal behavior policy based on which the generated experimental dataset yields the smallest semiparametric efficiency bound.
>   - **Optimal Design:** Theorem 1 derives the optimal behavior policy $\pi^{b*}$. For each day, $\pi^{b*}$ randomly allocates the initial action with probabilities proportional to $\sigma_*$,  the standard deviations of the cumulative rewards, and the same action is repeatedly assigned afterwards.
>   - **Implementation.** In real practice, $\sigma_*$ under the two global policies are unknown. We propose to use data collected so far to adaptively estimate $\sigma_*$. Specifically, we run each global policy for $m_0$ days to generate the data. Then we estimate $\sigma_*$ using the data collected so far and plug-in these estimators into $\pi^{b*}$ to generate the experimental data.
>
> * **Further improvement on the estimator.** Thank you for this insightful comment! The proposed ATE estimator and the doubly robust estimator by Kallus and Uehara (2020, https://dl.acm.org/doi/abs/10.5555/3455716.3455883) do exhibit differences.
>   - **Methodologically**, we estimate the ATE in an online manner (see Equation 4). One notable distinction is that the nuisance functions are computed during the data collection process using the data collected so far, and are independent of the data used to construct the policy value. Such a forward type cross-fitting procedure enables us to circumvent the need to impose certain metric entropy conditions (Diaz, 2020, https://academic.oup.com/biostatistics/article/21/2/353/5631845).  Meanwhile, it improves upon the doubly robust estimator in that our estimator's inherent design allows for online computations, negating the necessity to store all historical data.
>
>   - **Theoretically**, we delve into the statistical properties of the proposed online ATE estimator, establishing finite sample error bounds for its MSE.

---

> > ### Comment · Reviewer_bgAF · 2023-08-10
> > **Thank you!**
> >
> > Thank you very much for clarifying!

---

### Official Review · Reviewer_NJNM · 2023-07-04

**Soundness:** 4 excellent
**Presentation:** 4 excellent
**Contribution:** 2 fair
**Rating:** 6
**Confidence:** 5

**Summary:**

This paper studies optimal designs for allocating treatments so that treatment effects can be accurately estimated in online experiments. The focus of the paper is designing behavior policies for data generation to minimize the MSE of the ATE estimator when we know that dynamics may evolve through an NMDP, TMDP, or MDP.

**Strengths:**

1. The paper studies the challenging and important problem of SUTVA violations in online experiments. Without accounting for these violations, the temporal carryover effects that occur due to sequential allocation of treatment in online experiments can yield biased treatment effect estimates.
2. The authors consider an ambitious problem of designing optimal behavior policies when data generation evolves via NMDP, TMDP, or MDP.
3. Although there are many works that focus on off-policy evaluation and estimation of treatment effects, not many of these focus on how to design the behavior policy to improve the efficiency of the policy value estimator (it is more common to assume that a fixed dataset, generated according to some prespecified or unknown behavior policy, already exists).
4. The paper has a thorough and comprehensive related work section.
5. The paper is clear and well-written.


**Weaknesses:**

1. In Section 4, when analyzing the optimal designs for the TMDP and MDP models, the choice to restrict the treatment assignment policies that are considered to the class $\Pi^{b}$, the class of policies that randomly assign the first action and sticks with the same action for the rest of the trajectory, seems quite restrictive. In some sense, this removes some of the “dynamic” aspects of the original problem and greatly reduces the complexity of the problem.

2. It seems like the burn-in period is quite essential to estimating the ATE with the proposed designs because proposed behavior policies randomly selects an action $A_{1}$ in the first time step and then sticks with it for the remaining time steps. As a result, if we didn’t have the burn-in period, we wouldn’t be able to estimate both $V_{1}^{0}$ and $V_{1}^{1}$. The importance of the burn-in period isn’t emphasized in the text and is worth commenting on (or at least citing references that burn-in periods are necessary for consistent estimation).


**Questions:**

1. Can the authors discuss the choice to restrict the treatment assignment policies to class $\Pi^{b}$?
2. Is it possible to obtain good estimates of the ATE without requiring any burn-in period? (If not, it would be helpful to add a short comment on this to the paper.)
3. Is a burn-in period used in the experiments? If so, how long was the burn-in period? (Can this be added to the main text?)


**Limitations:**

Yes.

---

> ### Author Rebuttal · Authors · 2023-08-09
>
> * **Choice of class $\Pi^b$.**
>
>   - Many thanks for this constructive comment! First, we would like to clarify that it is an exceptionally challenging task to directly identify the optimal behavior policy $\{   \pi^b\_t \}\_t$ in TMDPs/MDPs that minimizes the following efficiency bound for estimating the ATE,
>     $$
>     \textrm{EB}\_2(\pi^b) =
>     		{1 \over T^2}
>     		\sum\_{t=1}^T \sum\_{a \in \{0, 1\} } \mathbb{E}^{\pi^b} \Big[ { \mathbb{I}(A\_t=a) p^a\_t ( O\_{t} )  \over p^b\_t ( O\_t,a )   } \sigma\_t( O\_t, a) \Big]^2+\frac{1}{T^2}\textrm{Var}[V\_1^1(O\_1)-V\_1^0(O\_1)],
>     $$
>     without any restrictions. The main challenge lies in that the marginal distribution function  $p^b_t ( O_t,a )$ cannot be conveniently expressed in a closed-form  function of $ \pi^b_t$ because it  depends on $\pi_t^b$ through a very complex mechanism. This introduces considerable difficulty in  obtaining the optimal behavior policy in a closed-form manner.
>
>   - Second, we clarify that we restrict the treatment assignment policies to $\Pi^b$ for several reasons:
>     (i) The optimal design for NMDP resides in class $\Pi^b$, and since both TMDP and MDP are subclasses of NMDP, we take advantage of this result and expect that the optimal in-class behavior policy that belongs to $\Pi^b$ performs well in TMDPs/MDPs as well; (ii) Theorem 3 demonstrates that under certain conditions, the optimal design does belong to $\Pi^b$, making the optimal in-class behavior policy globally optimal; (iii) Frequently alternating treatments can introduce significant carryover bias in policy evaluation, as noted in Hu and Wager, (2022) (https://arxiv.org/abs/2209.00197). Consequently, restricting our attentions to $\Pi^b$ allows us to avoid distributional shifts between the behavior policy and the target policy, effectively mitigating the carryover bias.
>
>   - Third, while it may not be possible to express the globally optimal behavior policy in closed-form, it can still be computed numerically, even without restricting to $\Pi^b$. For example, modern Bayesian optimal experimental design provides a powerful computational framework for optimizing the design of experiments (Blau et al., 2022, https://proceedings.mlr.press/v162/blau22a.html). However, this Bayesian approach is beyond the scope of our current proposal, which adopts a frequentist approach to derive the optimal design analytically. We will consider exploring the Bayesian direction in future work. We will add the related discussions shall our paper be accepted.
>
> * **The importance of burn-in period.**  This is an excellent suggestion! We totally agree that it would be undoubtedly helpful to discuss this point. During the rebuttal, we also conducted additional numerical experiments to empirically investigate the importance of the burn-in period. We will incorporate the related discussions, references and results shall our paper be accepted.
>
>   -  As you have commented, the burn-in period is used to derive reasonable initial estimators for the value function $V$ and the variance $\sigma^2$. If there is available historical data, these parameters could be estimated from that data, potentially bypassing the need for a burn-in period in a new experiment. However, in the absence of any prior information, the burn-in period becomes crucial for beginning the experiment with reliable initial estimates, as shown in our numerical experiments below. We also remark that such a burn-in procedure is widely used in experimental designs where the treatment assignment mechanism is initially unknown (Zhu et al. 2020, https://doi.org/10.1080/10543406.2019.1657439).
>   -  To empirically explore the impact of the burn-in period, we have conducted additional experiments under the settings described in Example 5.2 of the current paper.
>      We set $n=50$ and $T=10$. Table 2 of the attached pdf illustrates the Monte Carlo averages of the MSEs for the estimated ATE. Standard errors are provided in parentheses. Various burn-in periods are denoted by  $2m_0$. As the table indicates, smaller burn-in periods (e.g., $2m_0=4$) result in larger MSEs for the estimated ATE. This suggests that inaccurate initial estimators of $V$ and $\sigma^2$  can lead to unstable ATE estimators.  To the contrary, when the burn-in period is moderately large (e.g., $2m_0=16$ or $20$), the estimated ATE becomes more stable, with much smaller MSEs and standard errors. This experiment underscores the significance and influence of the initial burn-in duration.
>
> * **The burn-in period in the experiments.** Thanks for this question! In the supplementary material, we present  more details about the experiments. In particular, we set $m_0$ to $n /4 $ in all the experiments. We will mention this in Section 5 of the main text shall our paper be accepted.

---

> > ### Comment · Reviewer_NJNM · 2023-08-13
> >
> > Thank you for these clarifications! I hope the authors include some of these discussions in their final draft, and I recommend accepting this paper.

---

### Official Review · Reviewer_WpTu · 2023-07-04

**Soundness:** 4 excellent
**Presentation:** 4 excellent
**Contribution:** 3 good
**Rating:** 7
**Confidence:** 4

**Summary:**

The paper studies optimal treatment allocation aiming to maximize the obtained information from online experiments to estimate treatment effects accurately. The authors propose optimal allocation strategies in a dynamic setting. These strategies are designed to minimize the variance of the treatment effect estimator when data follow a NMDP or TMDP.

**Strengths:**

The problem that the paper investigates is important, in my opinion. The paper provides an insightful paradigm that how reinforcement learning techniques can be effectively applied in experimental design/causal inference. The authors also did a great job on giving credits to RL literature.

**Weaknesses:**

1.	There are several very relevant papers in experimental design literature that needs to be discussed carefully, for example, Bojinov et al. (2022) and Farias et al. (2022). I think some insights and the designs in this paper are closely related to these two papers.
2.	From experimental design/causal inference literature, we may not only want to know an estimator of ATE, but also the corresponding confidence intervals for the estimators. I am wondering that whether the authors could have some comments on how to construct confidence intervals.
3.	The definitions and relationships between $n$ and $T$ are a little bit confusing, especially when reading Section 2. I think some more descriptions on the relationship them will be helpful. From my understanding, the data structure looks very similar to “panel data” in the literature.

Reference:

Bojinov, I., Simchi-Levi, D., & Zhao, J. (2022). Design and analysis of switchback experiments. Management Science.

Farias, V., Li, A., Peng, T., & Zheng, A. (2022). Markovian interference in experiments. Advances in Neural Information Processing Systems, 35, 535-549.


**Questions:**

I have another minor question on the word “dynamic” in the title. Since the optimal allocation rule is $A_1=A_2=\cdots=A_T$, I am not pretty sure whether it is still suitable to call it a "dynamic" allocation.

**Limitations:**

See previous comments.

---

> ### Author Rebuttal · Authors · 2023-08-09
>
> * **Related Work on Experimental Design.**  Thank you for highlighting these pertinent references! Regrettably, they were overlooked in our initial manuscript submission. Should our paper be accepted, we will certainly incorporate and discuss these references. Specifically:
>
>   - Bojinov et al. (https://doi.org/10.1287/mnsc.2022.4583) delved into optimal design in the context of regular switchback experiments, operating under the assumption of a singular experimental unit. Their method employed the Horvitz-Thompson estimator, more widely recognized as the importance sampling estimator, to estimate causal effects. This technique presumes the presence of carryover effects in a fixed chronological sequence. Contrarily, our suggested experimental designs cater to NMDP and (T)MDP models incorporating multiple experimental units. Moreover, we investigate doubly robust estimators, which not only exhibit semiparametric efficiency but also offer greater resilience compared to the importance sampling estimator.
>
>   -Farias et al. (https://proceedings.neurips.cc/paper_files/paper/2022/hash/03a9a9c1e15850439653bb971a4ad4b3-Abstract-Conference.html) presented an on-policy estimator, specifically designed for off-policy evaluation in dynamical systems experiencing Markovian interference. Their main focus was the effective estimation of the average treatment effect (ATE) using a predefined set of offline data. In contrast, our research pursues a unique angle. We delve into the optimal experimental design, aiming to curate offline data that reduces the variance of the ATE estimator.
>
> * **Confidence Intervals of the ATE Estimator.**
>
>   - This is an excellent comment! We totally agree that establishing confidence intervals for the ATE estimators holds significant importance. As elaborated further below, the derivation of a confidence interval for the proposed estimator is quite straightforward. Should our paper be accepted, we will make sure to include these discussions.
>
>   - First, let us consider the settings under NMDPs in Section 3. Recall that after generating data through the proposed experimental design, we utilize the online doubly robust estimator $\widehat{\textrm{ATE}}_1$ to estimate the ATE. A key observation is that, under certain regularity conditions, the proposed estimator is asymptotically normal. More specifically, we have
>         $$\sqrt{n - 2m_0} (\widehat{\textrm{ATE}}_1 - \textrm{ATE}) \overset{d}{\rightarrow} N(0, \textrm{EB}_1(\pi^{b*})).$$  This motivates us to consider the following Wald-type confidence interval
>     $$ [\widehat{\textrm{ATE}}_1- \Phi^{-1} (1-\alpha/2) \sqrt{ \textrm{EB}_1(\pi^{b*})/(n - 2 m_0) },  \widehat{\textrm{ATE}}_1+ \Phi^{-1} (1-\alpha/2) \sqrt{ \textrm{EB}_1(\pi^{b*})/(n - 2 m_0) }],$$
>     where $\Phi^{-1}$ is the inverse cumulative distribution function of a standard normal random variable. It then suffices to estimate the asymptotic variance ${\textrm{EB}}\_1(\pi^{b*})$ to construct asymptotically valid confidence intervals. Notice that the estimate  can be represented as an average of martingale differences $\widehat{\textrm{ATE}}\_1= \sum_{i=2m_0 +1}^n \psi^1_i /(n - 2m_0)$  where $$ \psi\_i^1=\sum_{a=0}^1\frac{(-1)^{a+1}}{T} \Big[ \widehat{V}\_{1,i-1}^a(O_1^{(i)})+ \frac{\mathbb{I}(A\_1^{(i)}=a)}{\widehat{\pi}^{b*}\_{1,i-1}(a|O\_1^{(i)})}[\sum\_t R\_t^{(i)}-\widehat{V}\_{1,i-1}^a(O\_1^{(i)})]\Big]. $$ We propose using the sample variance of $\{\psi^1\_i\}\_i$ to estimate $\textrm{EB}\_1(\pi^{b*})$. Similar to Theorem 15 of Kallus and Uehara (2022) (https://dl.acm.org/doi/abs/10.5555/3455716.3455883), we can establish the consistency of the resulting sampling variance estimator.
>
>
>    - For TMDPs, we can similarly establish the asymptotic normality of $\widehat{\textrm{ATE}}_2$, i.e.,
>     $\sqrt{n - 2 m_0} ( \widehat{\textrm{ATE}}_2 -  \textrm{ATE} ) \overset{d}{\rightarrow} N(0, \textrm{EB}_2(\pi^{b*}) ) $. The corresponding $1 - \alpha $ confidence interval can be constructed by
>     $$
>     	[\widehat{\textrm{ATE}}_2- \Phi^{-1} (1-\alpha/2) \sqrt{ \textrm{EB}_2(\pi^{b*})/(n - 2 m_0) },  \widehat{\textrm{ATE}}_2+ \Phi^{-1} (1-\alpha/2) \sqrt{ \textrm{EB}_2(\pi^{b*})/(n - 2 m_0) }],
>     $$
>     where the unknown asymptotic variance $\textrm{EB}_2(\pi^{b*})$ can be similarly estimated via the sampling variance estimator.
>
>
> * **Definition and relationships between $n$ and $T$.**  We apologize for any potential confusions. You are correct! The data structure is similar to the panel data. We consider an experiment spanning $n$ days and each day is divided into $T$ time intervals. On the $i$-th day (i=1, $\dots$ , n) and time interval $t$ $(t=1, \dots, T)$,  the decision maker observes certain time-varying market features, denoted as $O_t^{(i)}$ (e.g., the numbers of incoming orders and available drivers in a ride-sharing platform) and chooses to implement one of the two policies. This action is denoted as $A_t^{(i)}\in \{0,1\}$. By convention, $A_t^{(i)}=1$ indicates the implementation of a new policy, while $A_t^{(i)}=0$ signifies the use of the control policy. Afterwards, the decision maker receives an immediate reward $R_t^{(i)}$ and subsequently observes the next observation $O_{t+1}^{(i)}$. The observed data can thus be summarized as $\{O_t^{(i)}, A_t^{(i)},  R_t^{(i)}, t=1, \dots, T\}_{i=1}^n$. We will add the related discussions, and make the definitions of $n$ and $T$ clearer shall our paper be accepted.
>
> * **Question about the title:** Thanks for your valuable suggestion! We will remove ``dynamic'' from our title shall the paper be accepted.

---

> > ### Author Response · Authors · 2023-08-10
> > **Thank you and Formulas Clarification**
> >
> > Dear Reviewer,
> >
> > We extend our heartfelt appreciation for your dedicated time and valuable insights. We are writing to address a technical issue we encountered during the submission process, which led to some of the formulas in our initial rebuttal not being displayed in the final version. We deeply regret any inconvenience this may have caused. In the following, we present a revised version of the rebuttal, specifically addressing the "Confidence Intervals of the ATE Estimator," for enhanced clarity.
> >
> > - **Confidence Intervals of the ATE Estimator.**
> >
> >   - This is an excellent comment! We totally agree that establishing confidence intervals for the ATE estimators holds significant importance. As elaborated further below, the derivation of a confidence interval for the proposed estimator is quite straightforward. Should our paper be accepted, we will make sure to include these discussions.
> >
> >   - First, let us consider the settings under NMDPs in Section 3. Recall that after generating data through the proposed experimental design, we utilize the online doubly robust estimator $\widehat{\textrm{ATE}}_1$ to estimate the ATE. A key observation is that, under certain regularity conditions, the proposed estimator is asymptotically normal. More specifically, we have
> >         $$\sqrt{n - 2m_0} (\widehat{\textrm{ATE}}_1 - \textrm{ATE}) \overset{d}{\rightarrow} N(0, \textrm{EB}_1(\pi^{b*})).$$
> >
> >      This motivates us to consider the following Wald-type confidence interval
> >     $$
> >     [\widehat{\textrm{ATE}}_1- \Phi^{-1} (1-\alpha/2) \sqrt{ \textrm{EB}_1(\pi^{b*})/(n - 2 m_0) },  \widehat{\textrm{ATE}}_1+ \Phi^{-1} (1-\alpha/2) \sqrt{ \textrm{EB}_1(\pi^{b*})/(n - 2 m_0) }],
> >     $$
> >
> >     where $\Phi^{-1}$ is the inverse cumulative distribution function of a standard normal random variable.
> >     It then suffices to estimate the asymptotic variance $\textrm{EB}\_1(\pi^{b*})$ to construct asymptotically valid confidence intervals. Notice that $\widehat{\textrm{ATE}}\_1$ can be represented as an average of martingale differences  $\widehat{\textrm{ATE}}\_1 = \sum_{i=2m_0 +1}^n \psi^1_i /(n - 2m_0)$ where
> >     $$
> >     \psi\_i^1=\sum\_{a=0}^1\frac{(-1)^{a+1}}{T} \Big[ \widehat{V}\_{1,i-1}^a(O\_1^{(i)})+ \frac{\mathbb{I}(A\_1^{(i)}=a)}{\widehat{\pi}^{b*}\_{1,i-1}(a|O\_1^{(i)})}[\sum\_t R\_t^{(i)}-\widehat{V}\_{1,i-1}^a(O\_1^{(i)})]\Big].
> >     $$
> >     We propose using the sample variance of $\\{\psi^1\_i\\}_i$ to estimate $\textrm{EB}_1(\pi^{b*})$. Similar to Theorem 15 of Kallus and Uehara (2022) (https://dl.acm.org/doi/abs/10.5555/3455716.3455883), we can establish the consistency of the resulting sampling variance estimator.
> >
> >   - For TMDPs, we can similarly establish the asymptotic normality of $\widehat{\textrm{ATE}}_2$, i.e.,
> >     $\sqrt{n - 2 m_0} ( \widehat{\textrm{ATE}}_2 -  \textrm{ATE} ) \overset{d}{\rightarrow} N(0, \textrm{EB}_2(\pi^{b*}) ) $. The corresponding $1 - \alpha $ confidence interval can be constructed by
> >     $$
> >     	[\widehat{\textrm{ATE}}_2- \Phi^{-1} (1-\alpha/2) \sqrt{ \textrm{EB}_2(\pi^{b*})/(n - 2 m_0) },  \widehat{\textrm{ATE}}_2+ \Phi^{-1} (1-\alpha/2) \sqrt{ \textrm{EB}_2(\pi^{b*})/(n - 2 m_0) }],
> >     $$
> >     where the unknown asymptotic variance $\textrm{EB}_2(\pi^{b*})$ can be similarly estimated via the sampling variance estimator.
> >
> > We are once again immensely grateful for your dedicated attention and constructive feedback.  In addition, if you have any additional questions or concerns, we would be glad to hear from you during the discussion period and provide clarification.

---

> > > ### Comment · Reviewer_WpTu · 2023-08-11
> > >
> > > Thank you very much for your extra efforts. The construction of the confidence interval, from my perspective, enhances the results of the paper a lot.

---

### Official Review · Reviewer_tKDd · 2023-07-12

**Soundness:** 3 good
**Presentation:** 4 excellent
**Contribution:** 3 good
**Rating:** 6
**Confidence:** 2

**Summary:**

This paper considers A/B tests which are often employed for evaluating new products/treatments/policies against existing baselines. The goal is to study the optimal design of such A/B tests to maximize the information obtained and estimate treatment effect more accurately. The paper considers three possible settings governing how the data is generated: whether it follows an MDP or if it follows a time-varying MDP or a non-Markov process and presents an optimal allocation strategy for each. The paper tests these policies on a real-world sourced dataset pertaining to a ride-sharing company and shows that these approaches achieve better accuracy. Finally, the paper also establishes nice theoretical properties of these methods and derive an upper boudn for the MSE of the proposed estimator.


**Strengths:**

– The paper studies an interesting problem that can be very useful in improving the efficiency of A/B tests that are commonly employed everywhere.

– Paper is well-written. It also sufficiently talks about existing related work and distinguishes itself.

– The key results in the paper are strongly grounded in theory and are supported by good results in the empirical section.


**Weaknesses:**

– Could there be some imbalance in allocation of treatment leading to fairness issues? For example, to optimize sample efficiency, could the algorithm withhold treatment unnecessarily if the placebo group has high variance? That way it could put most of the people in placebo and almost none in treatment? Or similarly, if there is a negative treatment but with high variance, it could opt to allocate that treatment more often?  It is unclear if the algorithm guards against these issues.

**Questions:**

– Please see concerns about potential weakness mentioned in above section. Additional question below:

– While the results look good in Fig 4, etc. could there be cases where errors on some of the points in proposed algorithms are really large, even though for most points they are lower than the baselines? For example, are there cases where these estimates become unstable leading to exploding errors?


**Limitations:**

No negative social impact (other than pointed out in questions section)

---

> ### Author Rebuttal · Authors · 2023-08-09
>
> * **Imbalance in Allocation.** This is an excellent comment. Indeed, a significant variance disparity between the two groups would inevitably lead to a marked imbalance in the treatment allocation. This is because in the design we proposed, the probabilities of treatment assignment are dictated by the ratio of the standard deviations of the two groups.
>   - **Imbalance unlikely to occur under weak signals**. However, in many real-world scenarios, the oracle treatment effect is relatively small. For example, in ridesharing platforms, the size of the treatment effect typically falls within a range of 0.5% to 2% (Tang et al., 2019, https://dl.acm.org/doi/pdf/10.1145/3292500.3330724). Given this modest treatment effect, it is reasonable to anticipate that introducing a new treatment will not drastically disrupt the system dynamics, resulting in outcomes with exceedingly high or low variances.
>   - **Burn-in diminishes the impact of imbalance**. Nevertheless, we understand the importance of tackling the issue of imbalance. To mitigate this problem, notice that our proposed algorithms consists of a burn-in period. During this period, both global policies are executed alternately over $2m_0$ days, independent of the variance ratio between the outcomes of the two groups. The impact of the imbalance is diminished as $m_0$ increases.
>   - **Additional approaches to address imbalance**. Meanwhile, there are other possible approaches to guard against the imbalance problem. One possible approach is to introduce a cutoff value for treatment allocation probabilities, avoiding extreme values and maintaining better balance. Specifically, truncation could be conducted when the estimated probability exceeds a certain threshold $p_1$ (e.g., $0.95$) or falls below another threshold $p_2$ (e.g., $0.05$). Furthermore, designating a minimum sample size, $\tilde{n}$, for each group beforehand can foster better balance. Should the sample size for one group exceed $n-\tilde{n}$, the subsequent days could be exclusively allocated to the other group, further reducing the risk of imbalance.
> * **Exploding Errors.**
>   - **Standard errors**. We greatly appreciate this helpful comment. In Tables 1-3 of the supplementary material, we have presented the MSEs of the estimated average treatment effect derived from our numerical studies along with their standard errors. By nature, a handful of significantly high MSEs can elevate the associated standard error. It can be seen from Tables 1-3 that our proposed method not only yield lower average MSEs, but also consistently produces smaller, or at the very least, comparable standard errors.
>   - **5-number summary statistics**. To provide a more comprehensive view, in Table 1 of the attached pdf file, we further give the 5-number summary statistics of the MSEs of the proposed estimators reported in Figure 4 of the main text. The findings from this table further reinforce that our proposed algorithms do not experience issues with exploding errors.

---

### Author Rebuttal · Authors · 2023-08-09

* We greatly appreciate all the reviewers' critical assessments of our work, many of which will lead to a more readable and self-contained version of our paper. We have worked very hard to address all comments and concerns, all discussions will be incorporated if our paper gets accepted.

* Rebuttals to specific comments are given in separate rebuttals to each review point by point. The attached pdf file contains two tables. Table 1 gives the 5-number summary statistics of the MSEs of the proposed estimators reported in Figure 4 of the main text. Table 2 reports the Monte Carlo averages of the MSEs with the estimated ATE, along with the corresponding standard errors in parentheses, with different burn-in periods.

* Thanks again for all the reviewers' insightful questions and feedback, which have helped us clarify and strengthen the foundations of our work. We hope our response clarified/addressed your questions about our paper.

---

### Decision · Program_Chairs · 2023-09-21

**Decision:**

Accept (poster)

**Comment:**

All reviewers recommend to accept this paper due to the following strengths:
* relevant and challenging problem
* clarity of the presentation

Despite initial concerns on the impact of the contributions, the reviewers believe they are significant after rebuttal and discussions. We encourage the authors to update their paper to reflect these discussions in their final version.